# Gas-particle partitioning of m-xylene and naphthalene oxidation products: temperature and NO$_x$ influence

Marwa Shahin[1], Julien Kammer[1], Brice Temime-Roussel[1], and Barbara D'Anna[1]

[1]Aix-Marseille Univ., CNRS, LCE, Marseille, France

*Correspondence to*: Barbara D'Anna (barbara.danna@univ-amu.fr), Marwa Shahin (marwa.shahin@etu.univ-amu.fr)

**Abstract.** Volatile organic compounds (VOCs) react with atmospheric oxidants resulting in oxygenated products of lower volatility known as semi and intermediate volatile organic compounds (S/IVOCs) forming secondary organic aerosols (SOA). Those compounds can partition between the gas and particle phases, a critical process influenced by several environmental parameters, yet poorly constrained. This study aims to evaluate the effect of temperature and VOC/NO$_x$ ratio on SOA formation and partitioning of individual SOA products from m-xylene and naphthalene OH-oxidation. Experiments are carried out in an oxidation flow reactor (OFR) and products are identified and quantified using a proton transfer reaction time-of-flight mass spectrometer (PTR-ToF-MS) coupled to a CHemical Analysis of aeRosol ONline (CHARON) inlet. Results show that lower temperatures significantly enhance SOA formation, while lower VOC/NO$_x$ ratios reduce it. Gas-phase m-xylene major products are C$_3$, C$_5$ and C$_8$ whereas particle-product distributions exhibit a progressive increase from C$_2$ to C$_8$. In contrast, naphthalene products partition more readily into the condensed phase, with C$_8$-C$_{10}$ products dominating. Most of the oxidation products from both precursors exhibit a volatility distribution in the SVOC regime, with fewer in the IVOC regime. The decrease in temperature shifts the effective saturation concentration ($C_i^*$) values towards lower values, though no clear relationship between $C_i^*$ and oxidation state is observed. A comparison between observed and estimated volatilities using a model based on group contribution method (SIMPOL.1) reveals systematic deviations for both light molecules and heavy compounds, suggesting a need for improved predictive models.

## 1 Introduction

Aromatic hydrocarbons (AHs) are an ubiquitous class of air pollutants and contribute to an important fraction of the total volatile organic compounds (VOCs) in urban environments; the relative contribution may vary and depends on location and season (Calvert et al., 2002; Jiang et al., 2017; Montero-Montoya et al., 2018). Among the different AHs, xylene and naphthalene are important anthropogenic VOCs primarily emitted from petrochemical industries, biomass burning, diesel and gasoline engines, or through solvent evaporation (Fang et al., 2021, 2024; Wu et al., 2020; Xuan et al., 2021). In the atmosphere, AHs react with common oxidants as hydroxyl (OH), nitrate (NO$_3$) and chloride (Cl) radicals and ozone (O$_3$) leading to the formation of oxygenated reaction products of lower vapor pressures also known as semi-volatile organic compounds (SVOCs) and intermediate-volatile organic compounds (IVOCs) (Seinfeld and Pandis, 2016). These compounds

may partition to the particle phase forming secondary organic aerosols (SOA) that represent approximately 60% of ambient organic aerosol (Huang et al., 2020, 2014) and have an impact on visibility (Li et al., 2014; Liu et al., 2017), climate (Liu and Matsui, 2020; Shrivastava et al., 2017) and human health (Anderson et al., 2012; Berlinger et al., 2024; Singh and Tripathi, 2021; Thangavel et al., 2022). Indeed, particulate air pollution is closely correlated to the progression of numerous respiratory diseases, in addition to cancer, cardiovascular diseases, and neurological damage (Liu et al., 2022a; Singh and Tripathi, 2021; Song et al., 2017; Thangavel et al., 2022). In the EU, each year, around 238,000 premature deaths are attributable to fine particulate matter ($PM_{2.5}$) exposure (European Environment Agency, 2022). Improving our understanding of particle sources and properties is thus pivotal to improve a more sustainable environment.

Over the last decades, many laboratory and modelling studies investigated SOA generated by AHs reaction products (Chen et al., 2018; Forstner et al., 1997; Klodt et al., 2023; Lannuque et al., 2018; Li et al., 2022; Liu et al., 2023, 2022b; Lu et al., 2024; Song et al., 2007; Srivastava et al., 2022; Tian et al., 2023). The OH radical oxidation of monoaromatic compounds operates through two main common pathways (Atkinson et al., 1991; Bloss et al., 2005; Calvert et al., 2002; Forstner et al., 1997; Pan and Wang, 2014). In the case of m-xylene, the first pathway is the OH radical addition to the aromatic ring, predominantly on the ortho position (Fan and Zhang, 2008) to form a hydroxy-methyl-benzyl radical with a subsequent addition of $O_2$ to form a bicyclic radical, a major ring opening product channel (Zhao et al., 2005). The second reaction pathway is the H-abstraction from the methyl group to form a methyl benzyl radical, with a subsequent addition of $O_2$ to form a benzyl peroxy radical (Atkinson et al., 1991; Molina et al., 1999). In the presence of $NO_x$, the peroxy radical mainly reacts with NO to form an alkoxy radical, and finally an abstraction by $O_2$ leads to the formation of m-tolualdehyde (Srivastava et al., 2023). In the case of naphthalene, OH radical reaction is initiated exclusively by OH addition, most predominantly at the α-carbon (68 %) adjacent to the fusion of its two aromatic rings (Wang et al., 2007). Then, a hydroxy cyclohexadienyl radical is formed, which further reacts with $NO_2$ to form nitrogen-containing compounds, or with $O_2$ to form various products out of which 2-formylcinnamaldehyde is major (Nishino et al., 2009).

The $NO_x$ level has been shown to have significant yet possibly contrasting effects on SOA formation (Chan et al., 2009; Liu et al., 2024; Ng et al., 2007; Qi et al., 2020; Sarrafzadeh et al., 2016; Song et al., 2005; Zhu et al., 2021). Some studies have reported a decrease in SOA yield under high $NO_x$ conditions which can be explained by the termination reactions of NO with the peroxy ($RO_2$ and $HO_2$) and OH radicals, limiting the formation of lower volatility compounds (Chan et al., 2009; Ng et al., 2007; Song et al., 2005). Nonetheless, Zhu et al. (2021) reported an increase in SOA mass under elevated levels of $NO_x$ compared to $NO_x$ free experiments. Enhanced SOA formation under high NOx has also been linked to the formation of organic nitrates and the isomerization of alkoxy radicals into low-volatility products, or related to a change in OH concentration due to the presence of $NO_x$ (Ng et al., 2007; Schwantes et al., 2019; Shi et al., 2022; Srivastava et al., 2023). Under low $NO_x$ conditions, $RO_2$ radicals primarily react with $HO_2$ to form low-volatility organic hydroperoxides, which contribute to new particle formation and increase SOA mass (Xu et al., 2014; Zhao et al., 2018). These findings illustrate that $NO_x$ is capable of both inhibiting and promoting SOA formation depending on the $NO_x$ regime.

A key process determining the fate of the oxidation products generated from AH oxidation is their ability to partition between gas and particle phases, either throughout nucleation forming new particles or by condensing on pre-existing particles. Partitioning is influenced by many environmental parameters such as temperature, relative humidity, the nature and the diameter of the pre-existing particles on which they condense and the physio-chemical properties of the S/IVOC condensing (Kamens et al., 2011; Kim et al., 2007; Lannuque et al., 2018; Loza et al., 2012; Ng et al., 2007; Qi et al., 2010; Sato et al., 2007; Takekawa et al., 2003; Warren et al., 2009; Xu et al., 2015).

It is well recognized that SOA yields increase at lower temperatures, a trend consistently reported for both terpenes and isoprene (Clark et al., 2016; Deng et al., 2021; Svendby et al., 2008; Virtanen et al., 2010), aromatics (Lannuque et al., 2023; Svendby et al., 2008), alkanes (n-dodecane in Fan et al., 2025; Li et al., 2020) and amines (Price et al., 2016). This behavior is attributed primarily to the decrease of the vapor pressures of the compounds, displacing the equilibrium towards the particle phase. However, the impact of temperature on SOA composition is not fully understood, and seems to depend on other experimental conditions (precursor, seed acidity, etc.). For biogenic precursors, previous studies reported more oligomer formation at lower temperatures (Li et al., 2020; Fan et al., 2025), driven by increased SVOC partitioning and condensed-phase reactions, while others observed the opposite, attributing higher-temperature oligomerization to radical or acid-catalyzed reactions (Clark et al., 2016; Deng et al., 2021; Price et al., 2016). Additionally, Li et al. (2019) highlighted how lower temperatures increase SOA viscosity, suppressing evaporation and favoring retention of low-volatility species. Conversely, Lamkaddam et al., (2017) found a weak temperature sensitivity in SOA formation from n-dodecane, suggesting that extremely low-volatility products and compensating shifts in product types may offset volatility effects. Regarding aromatic compounds, only Lannuque et al. (2023) investigated the effect of temperature on SOA chemical composition at molecular level, showing a general agreement of product distribution between 280 K and 295 K. Thus, more studies on different aromatic precursors and experimental conditions are needed to complete our understanding of the temperature effect on SOA formation from aromatic precursors.

Although temperature is recognized as a major factor governing gas-particle partitioning, most prior studies have mostly focused on the bulk SOA yields or selected classes of compounds (Ahn et al., 2021; Bahrami et al., 2024; John et al., 2018; Rutter and Schauer, 2007; Svendby et al., 2008; Takekawa et al., 2003; Wei et al., 2016; Zhang et al., 2023; Zhou, 2021). While these approaches have advanced the understanding of SOA formation and representation in models, they often lack molecular level resolved resolution. Experimental studies that probe the effect of temperature on the partitioning of individual SOA oxidation products, remain limited and relatively recent (Deng et al., 2021; Fan et al., 2025; Lannuque et al., 2023; Li et al., 2024). Such measurements are particularly important for refining SOA volatility parameterizations under atmospherically relevant cold conditions, such as those occurring during nighttime or wintertime episodes.

The oxidation flow reactor (OFR, Fig. 1) is a type of continuous-flow reactor that uses substantially elevated oxidant levels to rapidly simulate atmospheric oxidation chemistry (Kang et al., 2007; Peng and Jimenez, 2020). Lannuque et al. (2023) have studied the gas-particle partitioning of toluene in a similar OFR system. Experiments carried at 280 K were characterized by

a higher mass loading in the particle phase compared to those at 295 K, as well as a shift in volatility values of oxidation
products towards lower volatility values.
Direct measurements of S/IVOCs in real time remains challenging due to their complexity, diversity, and low concentrations.
The analytical development of the "CHemical Analysis of aeRosols ONline" (CHARON) inlet coupled with a proton-transfer-
reaction-time of flight-mass spectrometry (PTR-ToF-MS) enables alternative online measurements of both gas and particle
phases at molecular level (Eichler et al., 2015). It also allows measuring both gas and particle phases with reduced artefacts
associated with particle collection and thermal desorption compared to traditional techniques (Peng et al., 2023). Promising
applications have been shown in several laboratory and field studies (Lannuque et al., 2023; Müller et al., 2017; Muller et al.,
2017; Piel et al., 2021), while its application to gas-particle partitioning investigations is relatively new (Gkatzelis et al., 2018;
Lannuque et al., 2023; Peng et al., 2023; Piel et al., 2021).
The aim of the present study is to evaluate gas-particle partitioning of S/IVOCs involved in SOA formation from the
photooxidation of AHs in different conditions. For that purpose, the photooxidation of AH compounds was investigated using
an online CHARON-PTR-ToF-MS coupled to an oxidation flow reactor. These laboratory experiments aim at 1) identifying
the gas and particle phase products of m-xylene and naphthalene at a molecular level; 2) evaluating the partitioning behavior
of individual SOA products; and 3) assessing the effect of atmospheric conditions (temperature variation and $NO_x$/VOC ratio)
on this partitioning. Two compounds, m-xylene and naphthalene, are selected, for the following reasons, i) their ubiquity in
urban areas (Fang et al., 2021; Wu et al., 2020; Xuan et al., 2021) ii) their known reactivity with OH radicals that is in the
same order of magnitude (Calvert et al., 2015), and iii) their SOA formation potential has been previously demonstrated (Chan
et al., 2009; Chen et al., 2016, 2021; Kleindienst et al., 2012; Li et al., 2022; Loza et al., 2012; Lu et al., 2024; Ng et al., 2007;
Sato et al., 2022; Song et al., 2005, 2007; Srivastava et al., 2022; Takekawa et al., 2003; Ye et al., 2024; Zhang et al., 2019).
**2 Methods**
**2.1 The OFR experimental setup**
Photooxidation experiments have been conducted in a 19.3 L cylindrical aerosol oxidation flow reactor (OFR, 153 mm internal
diameter, 105 cm length) made up of quartz, vertically oriented, and surrounded by 6 UVB lamps (Helios Italquartz) with a
continuous emission spectrum in the 280-350 nm range ($\lambda_{max}$ = 310 nm, Fig. 1). An external air conditioning unit is connected
to the tube allowing temperature control in the range 280-295 K. The gas phase stream consists of humid air, the selected VOC
precursor (m-xylene or naphthalene), nitrogen dioxide ($NO_2$) and hydrogen peroxide ($H_2O_2$). For humidification, two glass
bottles containing milliQ water are bubbled at 0.2-1 L min$^{-1}$ by pure $N_2$ in order to maintain relative humidity (RH) around 50
% (from 35 % to 65 %, among all experiments). A constant concentration of m-xylene is generated by flowing a constant flow
of 0.15 L min$^{-1}$ of $N_2$ over a permeation tube containing a pure solution (≥ 99 % purity, Sigma Aldrich) placed in an oven and
kept at a constant temperature of 308 K. Some solid naphthalene (99% purity, Sigma Aldrich) is kept in an iced water bath
while headspace is flown at 0.1 L min$^{-1}$ by pure $N_2$ to generate a constant flow of naphthalene in the flow tube. $NO_2$ is
introduced using a cylinder ($100 \pm 5$ ppm in $N_2$, Linde) and diluted in $N_2$ using different mass flow controllers to the desired
mixing ratios prior to entering the tube (varying from 40 to 340 ppbV), depending on the required VOC/$NO_x$ conditions. A
hydrogen peroxide ($H_2O_2$) solution (50 % in $H_2O$, stabilized, Sigma Aldrich) is used as a hydroxyl radical (OH) precursor and
is constantly introduced in the OFR by bubbling pure $N_2$ in the solution at a flow of 0.1-0.2 L min$^{-1}$. For each experiment, the
OH radical concentration generated was estimated by fitting the VOC precursor (m-xylene or naphthalene) decay assuming a
pseudo first order reaction with OH radicals using temperature-dependent values of the kinetic rate constant as recommended
from NIST Kinetics Database. This estimation is also based on the hypothesis that the other reactions with OH do not limit the
reaction of the precursor in the first seconds following lamps switching on. The OH radical concentrations range from 2.2 to
$5.4 \times 10^7$ molecules cm$^{-3}$ corresponding approximately to 1.3 and 3 days of atmospheric OH-radical exposure, taking into
account a diurnal average hydroxyl radical concentration of $1.5 \times 10^6$ molecules cm$^{-3}$ (Mao et al., 2009). Due to background
contaminations, compounds with $m/z$ 61 and below, mainly acetaldehyde, acetic acid, formaldehyde and formic acid, are not
considered in further analysis. Monodispersed ammonium sulphate (AS) seeds serve as a pre-existing surface and are generated
by nebulizing a $10^{-2}$ M AS solution (99.5 % purity, Acros Organics) using a TSI atomizer (model 3076), dried through a silica
diffusion drier and then size selected in an Aerodynamic Aerosol Classifier (AAC, Cambustion) to generate monodisperse
aerosols with an average diameter of 200 nm. The overall input flow is 2.4 L min$^{-1}$ to ensure a residence time in the tube of 8
minutes. In this configuration, particle losses (or its transmissions through the OFR) were estimated by comparing the
concentration of seed particles at the inlet and outlet of the OFR, when generating seed at 200 nm electrical mobility diameters.
These losses were daily checked prior to each experiment and were in the range $10 \pm 5$ %. In addition, precursor losses were
estimated to be around 5 % for m-xylene and 10-15 % for naphthalene. Losses of gaseous products generated during SOA
experiments were not experimentally evaluated. Lannuque et al. (2023) showed, in a toluene SOA experiment, that wall losses
introduced a 10-15 % deviation on the SOA yield when considering both precursors and reactions products. For m-xylene, we
can reasonably assume lower losses as the residence time is shorter and the flow tube has a larger inner diameter. For
naphthalene, it is probable that the wall losses were higher than that of m-xylene.

## 2.2 Instrumentation and data analysis

Figure 1 shows the schematic of the experimental set-up, where several online instruments are used to characterize gas and
particle phase chemical compositions as well as particle number and size distribution.
A commercial Proton Transfer Reaction-Time of Flight-Mass Spectrometer (PTR-Tof-MS 6000X2, Ionicon Analytik GmbH,
Innsbruck, Austria) coupled to a "CHemical Analysis of aeRosols ONline" (CHARON) inlet is used to follow online the gas
and particle phase chemical composition of the organic fraction. The CHARON inlet has been already described in detail
elsewhere (Eichler et al., 2015; Leglise et al., 2019; Müller et al., 2017). Briefly, the sampled air travels through three major
sections of the CHARON inlet 1) a gas phase denuder of activated charcoal that strips off gaseous organics; 2) an aerodynamic
lens system (ADL) that collimates the subsampled flow and subsequently enriches the particle concentration and 3) a

thermodesorption unit (TD) heated at $T = 150°$ C that vaporizes the particles prior their introduction into the drift tube. The PTR-ToF-MS is used with hydronium ions ($H_3O^+$) to ionize organic analytes, and operates at a drift tube pressure of 2.6 mbar, a temperature of 120° C, and a voltage of 230 V. This results in an $E/N = 68$ Td ($E$ = electric field, $N$ = Number density of the gas molecules in the drift, 1 Td = $10^{-17}$ V cm$^2$ molecule$^{-1}$). It represents a relatively low $E/N$ compared to classical 120-140 Td reported in most of PTR-ToF-MS studies, as the aim is to minimize the potential fragmentation of parent ions and facilitate the molecular characterization of SOA. The potential higher dependence of the sensitivity to relative humidity variations at such low $E/N$ can be neglected as all experiments were conducted at fixed relative humidity (Pang, 2015; Tani et al., 2003). Instrument background has been daily performed using pure $N_2$, while sensitivity and particle enrichment factors (EF) are controlled at the end of the experiments. A blank was conducted prior to each SOA experiment, using the same conditions ($H_2O_2$ flow, NOx concentration, temperature, humidity, etc.), in the absence of the VOC precursor (either m-xylene or naphthalene). The products formed during these daily blanks were quantified and subtracted to the signal of the following experiment. EF is determined by CHARON calibration using a vanillic acid solution based on the method recommended by Eichler et al. (2015). Instrument sensitivity is evaluated by calculating the transmission curve using a cylinder containing 14 gas standards (benzene, toluene, ethylbenzene, o-, m-, p-xylene, styrene, 1,2,4-trimethyl-, 1,3,5-trimethylbenzene, chloro-, 1,2-dichloro-, 1,3-dichloro-, 1,4-dichloro-, trichloro benzene, each at $100 \pm 10\%$, ppb in $N_2$, RESTEK) covering a mass range up to $m/z$ 181.

IDA (Ionicon Data Analyzer 2.1.1.4) is used to process data recorded by the PTR-ToF-MS such as mass calibration, peak shape definition, peak identification and integration, rate constant calculation and VOC quantification (Müller et al., 2013). The peaks at $m/z$ 21.022 ($H_3^{18}O^+$), $m/z$ 330.847 corresponding to diiodobenzene ($C_6H_5I_2^+$) and its fragment at $m/z$ 203.943 ($C_6H_5I^+$) are used to recalibrate the mass scale (PerMaSCal® internal standard, 1,3-diiodobenzene, T = 55 °C). Molecular formulas composed of C, H, O, N atoms are assigned based on exact mass position, chemical rules (valence of atoms, for example) and isotopic patterns. The molecular formula is used to calculate dipole moments and polarizability as introduced by (Bosque and Sales, 2002) and (Sekimoto et al., 2017), which allows to calculate k-rate constants based on the ion-molecule collision theories (Gioumousis and Stevenson, 1958; Langevin, 1950; Su and Chesnavich, 1982). Concentrations are then estimated based on the rate constant between a proton and each VOC, the experimental transmission of each compound, and the primary ion intensity.

RStudio (RStudio 2023.06.0 Build 421) is used to perform a non-targeted approach for compound selection based on stability periods before and after oxidation. Time intervals are defined for each experiment corresponding to blank (photooxidation without VOC precursor, and HEPA (High-Efficiency Particulate Air) filter for particle phase), reactant (injection of products before the light has been turned on) and products (stable photooxidation), for both gas and particle phases. A Welch $t$-test is then used to statistically identify ions that are more concentrated than those in the blank. Then, among the selected ions, the products are defined as compounds more concentrated during oxidation period, and the invert for reactants, based also on Welch $t$-test.

A high-resolution Time-of-Flight Aerosol Mass Spectrometer (HR-ToF-AMS, Tofwerk AG, Aerodyne Inc. USA) is employed
for the quantification of the organic and inorganic aerosol fractions (Canagaratna et al., 2007; DeCarlo et al., 2006), with the
data recorded by the AMS being analyzed using the software SQUIRREL (ToF-AMS Analysis Toolkit 1.65C). The ionization
efficiency (IE) with respect to nitrate anions was $4.58 \times 10^{-8}$. It was calculated using nebulized 300 nm mobility diameter
ammonium nitrate particles (BFSP software). The relative IE (RIE) of ammonium was 3.45 based on the mass spectrum of
ammonium nitrate data from IE calibrations. The RIE of sulphate was determined by comparing the theoretical and the
measured concentration of a solution of ammonium nitrate and ammonium sulphate and was determined to be 1.75. For the
organic fraction, the default value of 1.4 was used. The AMS data were corrected by collection efficiency (CE) calculated by
comparison to the SMPS (Scanning Mobility Particle Sizer, TSI Classifier model 3082, DMA, TSI CPC 3776) volume using
densities of 1.7 g cm$^{-3}$ for ammonium sulphate and 1.4 g cm$^{-3}$ for organics. The CE values varied from 0.3 for pure ammonium
sulphate particles to 0.7 after SOA formation.
The SMPS is used for measuring particle size distribution and number concentration. Additional instruments account for a
chemiluminescent NO$_x$ analyzer (Envitec, model API200E), a pressure sensor (ATM.ECO, STS) and a temperature-humidity
probe (HMP9 Vaisala) positioned close the aerosol flow tube outlet, the last two instruments being connected to a data logger
(FieldLogger, NOVUS).

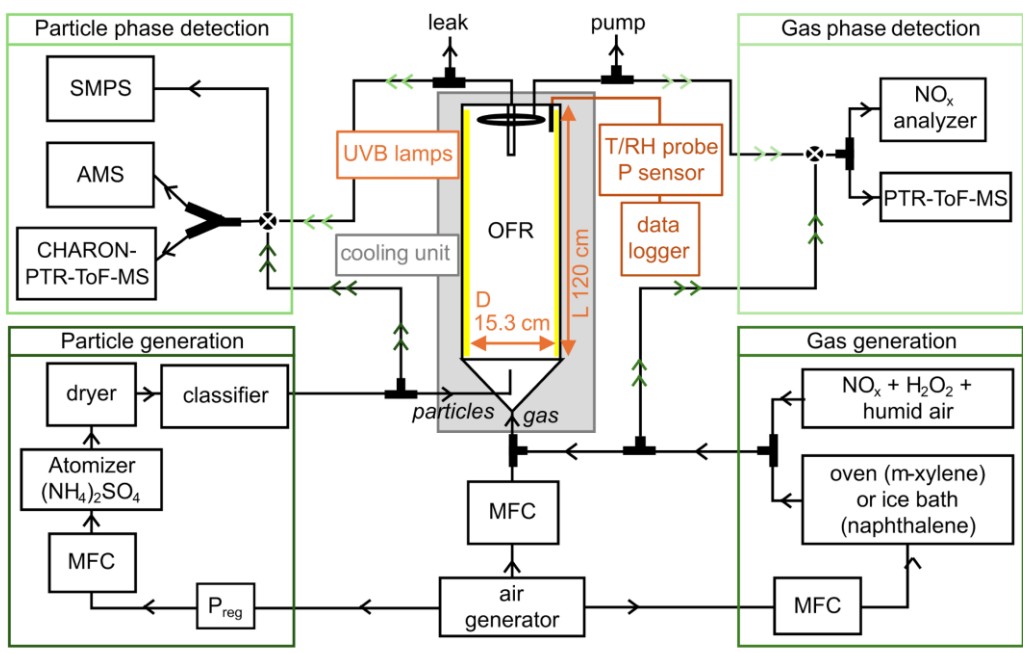

**Figure 1. Schematic of the experimental setup. MFC: mass flow controller.**

## 2.3 SOA yield, partitioning, and volatility distribution

The expression to describe the fractional aerosol yield ($Y$) was established by (Odum et al., 1996, 1997) and is described in Eq. (1), where $\Delta M_0$ is the amount of the total organic aerosol formed (in µg m$^{-3}$), and $\Delta VOC$ is the amount of VOC precursor reacted (in µg m$^{-3}$) measured as difference between inlet and outlet OFR.

$$Y = \frac{\Delta M_0}{\Delta VOC} \tag{1}$$

The distribution of oxidation products between the gas and particle phases can be explained by the partitioning theory of (Yamasaki et al., 1982):

$$K_{p,i} = \frac{C_{p,i}}{C_{g,i} \times TSP} \tag{2}$$

where $K_{p,i}$ is the experimental partitioning coefficient (in m$^3$ µg$^{-1}$) of a species $i$, $C_{p,i}$ and $C_{g,i}$ are concentrations (in µg m$^{-3}$) of the species $i$ in particle and gas phases, respectively, as measured by the CHARON-PTR-ToF-MS, and TSP is the total suspended particulate matter of the aerosol (in µg m$^{-3}$) as measured by SMPS. Larger $K_p$ values indicate a preference for a compound to partition in the particle phase. Subsequently, the volatility of the species can be defined by $\log_{10}C_i^*$, where $C_i^*$ is known as the effective saturation concentration (in µg m$^{-3}$) of a species $i$ which describes the gas-particle partitioning behavior of organic compounds and represents the gas-phase concentration of a compound at which it will partition equally between the gas and particle phases under given atmospheric conditions, and is calculated as the reciprocal of $K_{p,i}$ (Donahue et al., 2006, 2011):

$$C_i^* = \frac{1}{K_{p,i}} \tag{3}$$

The estimated values have been calculated using the Volcalc model based on molecular properties such as molecular weight, numbers of atoms and functional groups (Meredith et al., 2023; Riemer, 2023). The model is based on SIMPOL.1, a group contribution method (Pankow and Asher, 2008), which implements a structure-activity relationship method to calculate the subcooled pure liquid vapor pressure by summing the contributions of the subcooled liquid vapor pressures of individual chemical functional groups:

$$\log_{10}P_{L,i}^{\circ}(T) = \sum_k v_{k,i}b_k(T) \tag{4}$$

where $P^{\circ}_{L,i}(T)$ is the liquid vapor pressure (atm), $v_{k,i}$ is the number of groups of type $k$ in $i$, the index $k$ can take on the entire numbers (1,2,3, etc.), and $b_k(T)$ is the group contribution term for group $k$. No second-order interaction terms are included to account for neighboring functional groups, which means that the model does not consider the potential interactions or effects that adjacent functional groups might have on each other but only sums the contributions of individual functional groups independently.

The saturation concentration ($C^{\circ}_{i,T}$) of the major identified species $i$ is calculated at T = 280 and 295 K as follows:

$$C^{\circ}_{i,T} = C^{\circ}_{i,293} \times \frac{293}{T} \times exp\,exp\left(\left(\frac{-\Delta H}{R}\right) \times \left(\frac{1}{T} - \frac{1}{293}\right)\right) \tag{5}$$

where $C^\circ_{i,293}$ is the saturation concentration calculated by Volcalc at T = 293 K, $\Delta H_i$ is the enthalpy of vaporization of species $i$ (computationally predicted values from ChemSpider), and R is the ideal gas constant = 8.314 J mol$^{-1}$ K$^{-1}$.

The saturation concentration ($C_i^*$) and the effective saturation concentration ($C_i^*$) are related through the activity coefficient ($\gamma_i$) that captures the non-ideal interactions of the compound with the aerosol mixture. Its value generally lies between 0.3 (readily partitions to particle phase) and 3 (readily partitions to gas phase) for ambient atmospheric aerosol (Donahue et al., 2011; Liu et al., 2021). As in previous studies, such as Isaacman-VanWertz et al. (2016) and Nie et al. (2022), in this work we assume a $\gamma$ value of 1.

## 3 Results and discussion

### 3.1 SOA yield formation

Table 1 summarizes experimental conditions, consumed reactants ($\Delta VOC$) and formed organic aerosol mass ($\Delta M_0$) used to calculate the SOA yield ($Y$) for each experiment (Eq. (1)). Figure 2 presents SOA yields for (a) m-xylene and (b) naphthalene, comparing them with results from selected previous studies (Chan et al., 2009; Chen et al., 2018; Ng et al., 2007; Song et al., 2005). Filled markers indicate high NO$_x$ conditions (VOC/NO$_x$ < 8), empty markers refer to low NO$_x$ conditions (VOC/NO$_x$ > 8) (Dodge, 1977; NARSTO. and Electric Power Research Institute., 2000). The red square and blue triangle markers refer to this study at 295 K and 280 K, respectively, while all other experiments are conducted at room temperatures (between 295 K and 300 K).

**Table 1. List of conducted laboratory experiments and associated conditions, such as OFR temperature, RH, VOC/NO$_x$ ratio, seeds mass and SOA yield.**

| | T | RH | VOC | NO$_x$ | VOC/NO$_x$ | Seeds | $[OH] \times 10^7$ | $\Delta VOC$ | | $\Delta M_0$ | Y |
|---|---|---|---|---|---|---|---|---|---|---|---|
| | K | % | ppbV | ppb | ppbC ppb$^{-1}$ | µg m$^{-3}$ | molecules cm$^{-3}$ | µg m$^{-3}$ | % | µg m$^{-3}$ | % |
| **m-xylene** | $280 \pm 1.5$ | $75 \pm 5$ | $74 \pm 0.65$ | 235 | $2.5 \pm 0.1$ | $51 \pm 1.6$ | $3.4 \pm 0.5$ | $114 \pm 3.3$ | 34 | $26.4 \pm 1.2$ | $23.1 \pm 1.2$ |
| | $280 \pm 1.5$ | $50 \pm 5$ | $69 \pm 0.70$ | 40 | $13.9 \pm 0.5$ | $35 \pm 0.5$ | $2.2 \pm 0.4$ | $95 \pm 3.3$ | 30 | $26.1 \pm 1.9$ | $27.5 \pm 2.1$ |
| | $295 \pm 2$ | $60 \pm 7$ | $73 \pm 0.66$ | 221 | $2.6 \pm 0.1$ | $30 \pm 0.8$ | $5.4 \pm 0.8$ | $155 \pm 3.2$ | 49 | $12.6 \pm 1.4$ | $8.1 \pm 1.0$ |
| | $295 \pm 2$ | $55 \pm 7$ | $83 \pm 0.74$ | 40 | $16.6 \pm 0.6$ | $34 \pm 0.7$ | $4.7 \pm 0.7$ | $153 \pm 3.2$ | 42 | $21.1 \pm 1.4$ | $13.8 \pm 1.0$ |
| **naphthalene** | $280 \pm 1.5$ | $40 \pm 3$ | $57 \pm 0.35$ | 340 | $1.7 \pm 0.1$ | $46 \pm 0.6$ | $2.9 \pm 0.2$ | $92 \pm 2.3$ | 29 | $23.3 \pm 1.8$ | $25.3 \pm 2.0$ |
| | $280 \pm 1.5$ | $35 \pm 3$ | $53 \pm 0.41$ | 62 | $9.3 \pm 0.4$ | $36 \pm 1.0$ | $3.2 \pm 0.3$ | $79 \pm 2.3$ | 27 | $33.6 \pm 6.9$ | $42.7 \pm 8.8$ |
| | $295 \pm 2$ | $40 \pm 5$ | $53 \pm 0.47$ | 340 | $1.6 \pm 0.1$ | $55 \pm 1.5$ | $3.5 \pm 0.3$ | $84 \pm 2.3$ | 28 | $12.6 \pm 1.7$ | $14.9 \pm 2.1$ |

| | | | | | | | | | | | |
|---|---|---|---|---|---|---|---|---|---|---|---|
| $295 \pm 2$ | $50 \pm 5$ | $49 \pm 0.41$ | 57 | $8.6 \pm 0.4$ | $65 \pm 0.3$ | $3.1 \pm 0.4$ | $75 \pm 2.2$ | 29 | $13.8 \pm 1.6$ | $18.3 \pm 2.2$ |

259

For m-xylene (Fig. 2a), the SOA yield at 295 K under high NOx conditions is approximately 8 % (filled red square), in line with values reported by Ng et al. (2007) and Chen et al. (2018). Chen et al. (2018) reported yields with higher VOC/NO$_x$ ratios, slightly lower precursor concentrations (44-59 ppb), and without using seeds. The absence of seed particles reduces the available surface area for condensation of oxidized products (Lambe et al., 2015). Ng et al. (2007) conducted experiments with AS seeds as in our study, but using nitrous acid (HONO) as OH radical precursor, with varying initial xylene concentrations (from 42 to 171 ppb) which may explain SOA yield variability (from 3 to 8 %). The effect of higher initial levels of the VOC precursor has been shown to reduce aerosol formation (and amount of reacted precursor) and the formation of oxidation products, especially those with m/z > 110 da (Chen et al., 2019). It may be due to the competition between IVOC reactions with OH that produces LVOC/ELVOC, and precursor oxidation. In our case, the initial m-xylene concentration was kept similar between experiments to isolate the effect of temperature and NO$_x$ on SOA yield and chemistry. Both studies by Ng (2007) and Chen (2018) were conducted in dry conditions. Li et al. (2022) reported SOA yields increasing from 6.3 to 14 % when humidity increased from 10 % to 70 %. The order of magnitude of SOA yields in this study is close to what we observed, the variability of yields being associated mostly to NO$_x$ or NH$_3$ initial concentrations. Under low NO$_x$, a SOA yield of 14 % (empty red square) is observed, lying on the lower level (12-30 %) of reported values by Song et al. (2007), that used lower levels of xylene (39-52 ppb).

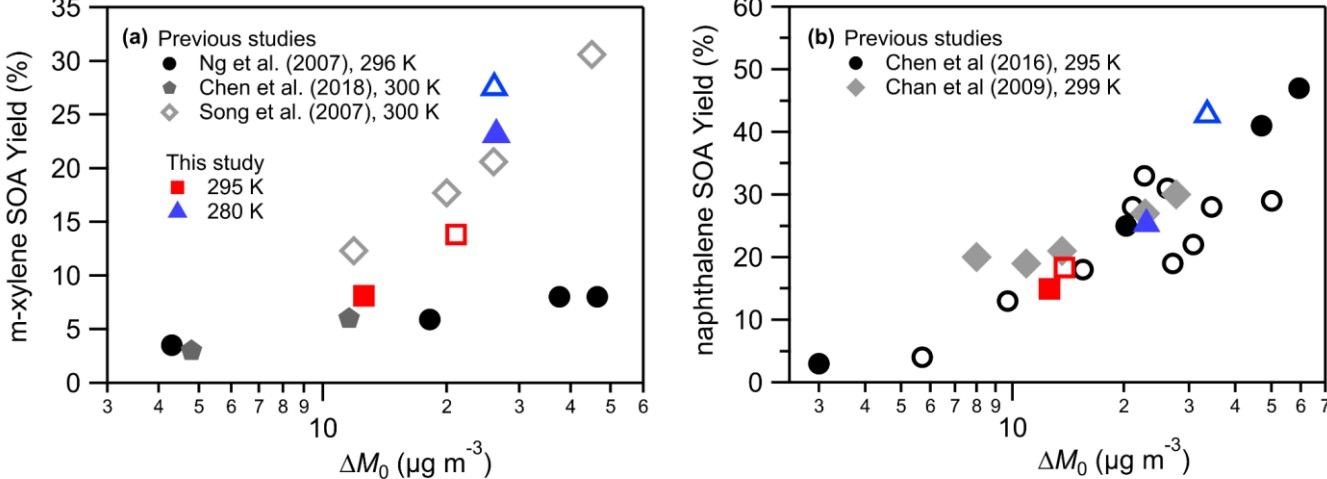

**Figure 2. SOA yields at 295 K and 280 K as function of organic aerosol mass formed for (a) m-xylene and (b) naphthalene in comparison with previous studies. Filled markers correspond to high NO$_x$ conditions, open markers to low NO$_x$.**

The effect of temperature on SOA yield is also depicted in Fig. 2. When decreasing from 295 K to 280 K, SOA yield increases from 8 to 23 % under high NO$_x$ conditions and from 14 to 28 % under low NO$_x$ conditions. Only a few studies investigated

the role of low temperature on SOA formation from monoaromatic precursors. Regarding xylene, only Takekawa et al. (2003) reported that SOA yield is enhanced by a factor around 2 (from 6 % to 13 %, on average) when temperature decreases from 303 K to 283 K at a VOC/NO$_x$ around 10 (like low NO$_x$ conditions in this study). This is in good agreement with our results, where a decrease of 15 K doubles the SOA yield under low NO$_x$ conditions (Fig. 2). Lannuque et al., (2023) found a similar temperature dependence of toluene-SOA and highlighted that the effect was greater for low concentrations of the precursors.

For naphthalene (Fig. 2b), SOA yield at 295 K with high NO$_x$ conditions ranges from 15 to 18 %, in agreement with Chen et al. (2016) and Chan et al. (2009) that reported values from 3 % to 47 %. Chen et al. (2016) observed the lowest SOA yield under high NO$_x$, attributed to the lowest amount of OH generated and highest NO injection. Even though Chan et al. (2009) used AS seeds, the initial amount of naphthalene was less than half of that used in our experiments and the humidity was below 10 %, highlighting again the important effect of experimental conditions on the prediction of SOA yields. Lower naphthalene SOA yields are observed under high NO$_x$ conditions (filled vs empty square or triangle, Fig. 2). Under low NO$_x$ conditions, a yield of 18 % is observed in quite good agreement with Chen et al. (2016), who reported yields varying from 4 % to 29 % as a function of the VOC/NO$_x$ ratio, with lowest yields for highest NO$_x$ regimes. A consistent increase in SOA yield is observed when switching from high to low NO$_x$ regime at 280 K, from 25 to 43 % at 280 K, while at 295 K the variation is limited to a few percent increase. The presence of NO$_x$ in the system may promote a competition for the termination reaction of RO$_2$ radicals between peroxy radicals (RO$_2$ and HO$_2$) and NO$_x$ (NO$_2$ and NO), leading to less oxidized products of higher volatility (Henze et al., 2008; Kroll and Seinfeld, 2008).

The effect of temperature on naphthalene SOA is reported here for the first time. Reducing the temperature from 295 K to 280 K induces an increase from 14.9 % to 25.3 % for high NO$_x$ conditions, and from 18.3 % to 42.7 % for low NO$_x$ conditions. SOA yields from naphthalene are in general higher compared to m-xylene, which is expected considering larger carbon skeleton of naphthalene (Aumont et al., 2013; La et al., 2016), and are in good agreement with previous studies (Chan et al., 2009; Chen et al., 2018). However, the effect of temperature on naphthalene SOA yield is slightly lower compared to the m-xylene SOA system. This may be the result of the larger carbon skeleton of naphthalene, leading to products of lower volatility, that have already partitioned more in the particle phase at 295 K.

## 3.2 Chemical composition of oxidation products in gas and particle phases

### 3.2.1 m-xylene

Figure 3 presents the chemical distribution of gas (Figs. 3a,c) and particles (Figs. 3b,d) phase products as a function of the carbon atoms following m-xylene oxidation by OH radicals. The mass fraction (in µg m$^{-3}$) is further sub-classified based on the number of oxygen (Figs. 3a,b) and nitrogen atoms (Figs. 3c,d; according to color scale), and the molecular weight distribution was divided into three groups: $m/z$ below 100, $m/z$ 101-150, and $m/z$ above 150. The experiment has been carried out under high NO$_x$ at 280 K. The total m-xylene carbon balance varies within 26-48 % for the detected gas phase products depending on the oxidation conditions and considering that CO, CO$_2$ and glyoxal are not measured. The latter compound has

strong interferences with the high signal of acetone. The SOA formed corresponds to 2.5 % of the total carbon balance at
295 K and 7 % at 280 K under high NO$_x$ conditions.

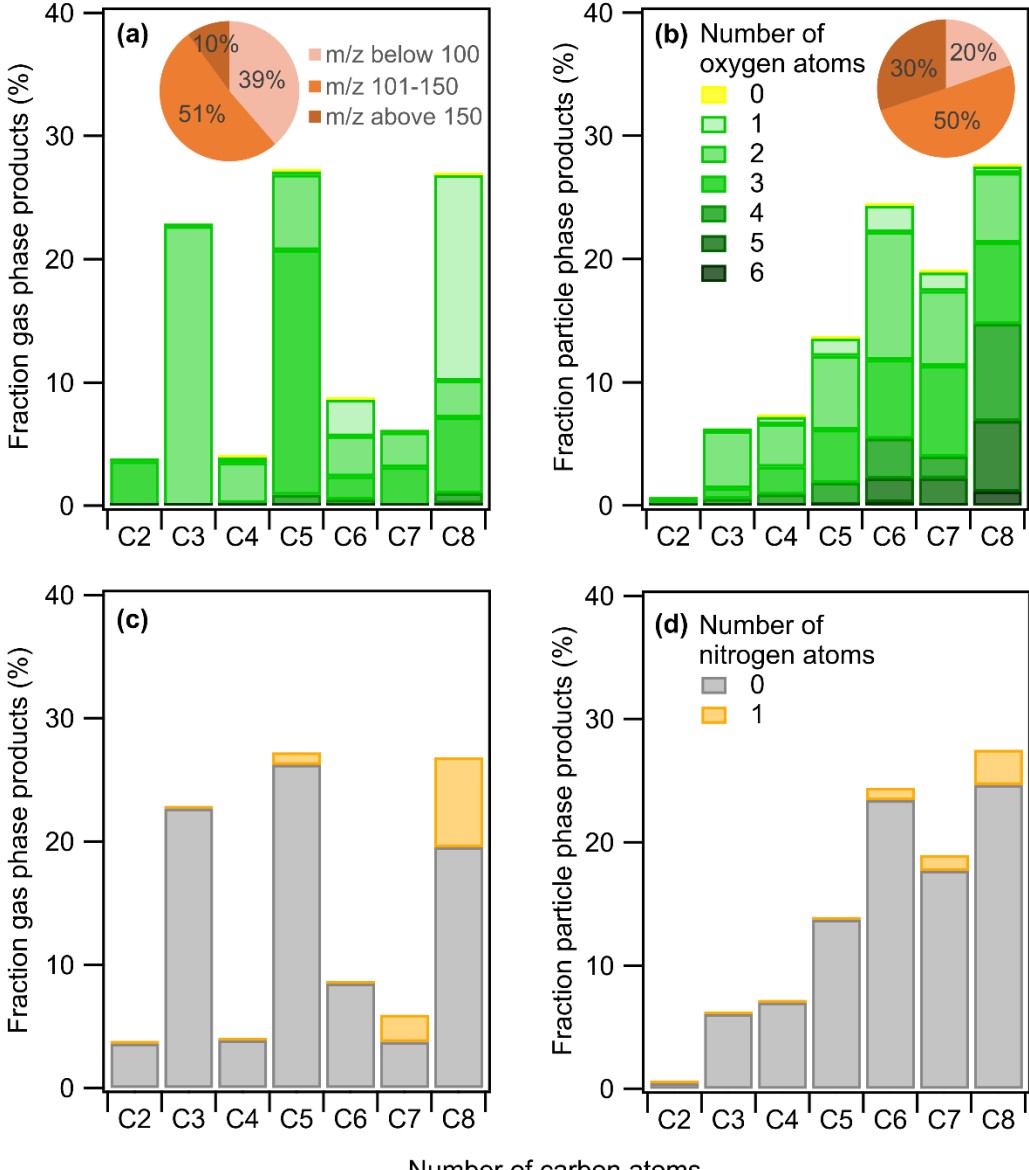


**Figure 3. m-xylene mass products fraction (y-axis) distribution based on the number of carbon atoms (x-axis) for a high NO$_x$**
**experiment at 280 K, colored by the number of (a,b) oxygen and (c,d) nitrogen atoms. Detected compounds are in the (a,c) gas phase**
**and (b,d) particle phase. Pie charts correspond to the molecular weight contribution to the overall mass.**
The overall product distribution at 295 K is similar to that at 280 K and can be found in the SI (Figs. S1a and S1b). At both
temperatures, the gas phase is mainly composed of C$_3$, C$_5$ and C$_8$ products with 1 to 3 O atoms for most of them, while the

particle phase is dominated by $C_6$, $C_7$ and $C_8$ compounds, containing generally 2 to 5 O atoms. In the gas phase, $C_8$ compounds are ring retaining compounds related to first-generation products, while $C_3$ and $C_5$ products arise from a more advanced chemistry. A larger fraction of $C_3$ products is present in the gas phase at 295 K when compared to 280 K (Figs. 3 and S1a). This could be due to the higher OH concentration for the experiment at 295 K (Table 1), leading to a higher consumption of $C_6$ and $C_8$ oxidation products due to their faster reactivity. This is supported by the time evolution of some compounds such as $C_6H_8O$ at $m/z$ 97.06, $C_6H_6O_2$ at $m/z$ 111.04, $C_6H_8O_4$ at $m/z$ 145.05, $C_8H_{10}O_5$ at $m/z$ 187.06 (Fig. S3). This hypothesis is also backed up by previous studies reporting the reaction of oxidation products (Cappa and Wilson, 2012; Isaacman-VanWertz et al., 2018; Jin et al., 2021). While for experiments with lower OH concentration, as at 280 K, the gas phase products (Fig. 3a) are characterized by a higher fraction of $C_8$ (26%) and $C_5$ (27%) compounds.

The condensed phase is enriched in compounds with higher molecular weight when compared to the gas phase, along with the presence of four or more oxygen atoms which highly reduces their vapor pressure (Cappa and Wilson, 2012). Particle phase products generally exhibit an increasing trend from $C_1$ to $C_8$ (Fig. 3b), similar to a recent study on toluene SOA (Lannuque et al., 2023).

The presence of $NO_x$ may affect product distribution, directly through the formation of organic nitrates, or indirectly by playing a role in shaping oxidation chemistry depending on experimental conditions. The comparison of the chemical composition of gas phase products in low $NO_x$ and high $NO_x$ at 295 K showed a general similar distribution (Fig. S1 and S2). The only noticeable change is the stronger contribution of $C_3$ and $C_5$ compounds in the low $NO_x$ experiment resulting from the slight increase in OH radical production as described above (Table 1), rather than a real $NO_x$ effect. This is supported by the same product distribution in particle phase, when comparing high and low $NO_x$, with only a small increase in organic nitrates at high $NO_x$. The very close distribution between high and low $NO_x$ means that the chemistry was not strongly dependent on the $NO_x$ regime in our conditions, and that $NO_x$ did not significantly affect the oxidant levels in the OFR.

Table 2 lists identified ions from the gas and particle phase products for two experiments at high $NO_x$ at 295 K and 280 K. The columns present the measured $m/z$ signals, the attributed chemical formula and a tentative chemical assignment based on results from previous studies on m-xylene photooxidation (Atkinson et al., 1991; Forstner et al., 1997; Huang et al., 2008; Jang and Kamens, 2001; Li et al., 2018, 2022; Zhang et al., 2019a, b; Zhao et al., 2005) and on the CHARON assignment procedure recommended by Gkatzelis et al. (2018) where average saturation mass concentration $\log_{10}(C_i^*)$ are used to discriminate between parent and fragment ions. The compounds are sorted in decreasing numbers of carbon atoms. For each temperature, two columns are presented: the first indicates the contribution of each compound to total gas phase organic products and the second the contribution to SOA.

**Table 2. Ions and their corresponding formulas of the major m-xylene products detected during photooxidation experiments under high $NO_x$. Reaction products are given as a fraction of the gas phase products (in % of µg m$^{-3}$) and as a fraction of the SOA products (in % of µg m$^{-3}$).**

| Carbon number | Measured $m/z$ and ion sum formula | Tentative assignment | $T$ = 298 K | | $T$ = 280 K | |
|---|---|---|---|---|---|---|
| | | | Gaseous | SOA | Gaseous | SOA |

| | | | Products (%) | Products (%) | Products (%) | products (%) |
|---|---|---|---|---|---|---|
| 8 | 121.06 (C$_8$H$_8$O)H$^+$ | m-tolualdehyde | 13.9 | 0.6 | 16.7 | 0.5 |
| 8 | 155.07 (C$_8$H$_{10}$O$_3$)H$^+$ | dimethyl-epoxy-oxo-hexenal/ trihydroxy dimethyl benzene | n.d. | 3.8 | n.d. | 3.3 |
| 8 | 137.06 (C$_8$H$_8$O$_2$)H$^+$ | toluic acid and possible contribution frag. 155.07 | n.d. | 8.5 | n.d. | 5.7 |
| 8 | 171.07 (C$_8$H$_{10}$O$_4$)H$^+$ | dihydroxy-dimethyl-cyclohexenedione / dimethyl-hexadienedioic acid | 0.3 | 4.6 | 0.3 | 5.6 |
| 8 | 153.06 (C$_8$H$_8$O$_3$)H$^+$ | hydroxy dimethyl quinone | 1.9 | 2.8 | 2.4 | 2.9 |
| 8 | 187.06 (C$_8$H$_{10}$O$_5$)H$^+$ | hydroxy-cyclohexene-dicarboxylic acid oxo-cyclohexanedicarboxylic acid | 0.1 | 2.6 | 0.1 | 2.5 |
| 8 | 169.05 (C$_8$H$_8$O$_4$)H$^+$ | dihydroxy-methylbenzoic acid | n.d. | 2.0 | n.d. | 1.6 |
| 8 | 185.05 (C$_8$H$_8$O$_5$)H$^+$ | trihydroxy-(hydroxymethyl)benzaldehyde | n.d. | 1.0 | n.d. | 0.7 |
| 8 | 168.06 (C$_8$H$_9$NO$_3$)H$^+$ | *dimethyl nitrophenol* | 1.9 | 0.4 | 3.6 | 0.4 |
| 8 | 152.07 (C$_8$H$_9$NO$_2$)H$^+$ | *nitro-xylene* | 0.4 | 0.1 | 1.8 | 0.1 |
| 7 | 125.06 (C$_7$H$_8$O$_2$)H$^+$ | dimethyl-pyranone/ methyl-hexadienedial | n.d. | 3.6 | 1.6 | 5.0 |
| 7 | 141.05 (C$_7$H$_8$O$_3$)H$^+$ | methyl-oxo-hexadienoic acid /heptenetrione | 0.7 | 2.8 | 0.7 | 4.0 |
| 7 | 139.04 (C$_7$H$_6$O$_3$)H$^+$ | hydroxy benzoic acid/ hydroxy methyl benzoquinone | n.d. | 3.1 | n.d. | 2.4 |
| 7 | 157.05 (C$_7$H$_8$O$_4$)H$^+$ | hydroxy-dioxo-heptenal / epoxymethylhexenedial | n.d. | 2.0 | n.d. | 1.8 |
| 7 | 109.06 (C$_7$H$_8$O)H$^+$ | cresols/ benzyl alcohol | n.d. | 1.4 | n.d. | 1.5 |
| 7 | 173.06 (C$_7$H$_8$O$_5$)H$^+$ | hydroxy-dioxo-heptenoic acid | n.d. | 1.8 | n.d. | 1.2 |
| 7 | 138.06 (C$_7$H$_7$NO$_2$)H$^+$ | *nitrotoluene* | 1.1 | 0.4 | 1.2 | 0.3 |
| 7 | 154.05 (C$_7$H$_7$NO$_3$)H$^+$ | *nitrocresol* | 0.7 | 0.3 | 0.6 | 0.3 |
| 6 | 127.04 (C$_6$H$_6$O$_3$)H$^+$ | hydroxymethyl furfural/ hydroxyquinol/dimethylfurandione | 1.0 | 2.4 | 1.4 | 3.3 |
| 6 | 113.06 (C$_6$H$_8$O$_2$)H$^+$ | methyl-oxo-pentenal / dimethylfuranone | n.d. | 1.7 | 0.2 | 2.2 |
| 6 | 129.06 (C$_6$H$_8$O$_3$)H$^+$ | hydroxy-oxo-hexenal methyl-oxo-pentenoic acid | 0.4 | 1.6 | 0.4 | 2.7 |
| 6 | 111.04 (C$_6$H$_6$O$_2$)H$^+$ | methylfuraldehyde / benzenediols possible frag. of 129.06 | n.d. | 3.6 | 2.7 | 6.2 |
| 6 | 143.03 (C$_6$H$_6$O$_4$)H$^+$ | dioxo-hexenoic acid/ methyl-dioxo-pentenoic acid tetrahydroxybenzene | n.d. | 1.4 | 0.2 | 1.8 |
| 6 | 159.04 (C$_6$H$_6$O$_5$)H$^+$ | oxo-hexenedioic acid | n.d. | 1.2 | n.d. | 1.2 |
| 6 | 145.05 (C$_6$H$_8$O$_4$)H$^+$ | hydroxy-dioxo hexanal | n.d. | 1.2 | 0.1 | 1.1 |
| 6 | 115.07 (C$_6$H$_{10}$O$_2$)H$^+$ | cyclopentylcarboxylic acid | n.d. | 0.4 | 0.4 | 1.1 |
| 6 | 95.03 (C$_6$H$_6$O)H$^+$ | phenol | n.d. | 1.4 | n.d. | 1.2 |
| 5 | 113.02 (C$_5$H$_4$O$_3$)H$^+$ | methyl-furandione | 16.7 | 1.3 | 19.3 | 2.1 |
| 5 | 131.04 (C$_5$H$_6$O$_4$)H$^+$ | methy-hydroxy-oxo-butandial | 2.1 | 1.4 | 0.9 | 0.9 |
| 5 | 117.05 (C$_5$H$_8$O$_3$)H$^+$ | oxo-pentanoic acid + frag at 99.04 | 1.2 | 0.7 | 0.2 | 0.4 |
| | 99.04 (C$_5$H$_6$O$_2$)H$^+$ | oxo-pentenal / methyl-butendial | 6.7 | 3.5 | 2.8 | 3.9 |
| 5 | 115.03 (C$_5$H$_6$O$_3$)H$^+$ | oxo-pentenoic acid | 1.4 | 2.4 | 0.7 | 1.8 |
| 5 | 97.03 (C$_5$H$_4$O$_2$)H$^+$ | furaldehyde | 1.1 | 0.7 | 2.2 | 1.7 |

| | | | | | | |
|---|---|---|---|---|---|---|
| 5 | 101.06 ($C_5H_8O_2$)H$^+$ | oxo-pentanal and isomers + frag at 83.05 | n.d | 0.4 | n.d | 0.4 |
| | 83.05 ($C_5H_6O$)H$^+$ | methylfuran | 0.6 | 1.0 | 0.3 | 1.3 |
| 4 | 71.05 ($C_4H_6O$)H$^+$ | dihydrofuran / MACR / MVK | 1.1 | 0.3 | 0.4 | 0.3 |
| 4 | 87.04 ($C_4H_6O_2$)H$^+$ | butanedial / crotonic acid | n.d. | 2.0 | n.d. | 2.2 |
| 4 | 103.04 ($C_4H_6O_3$)H$^+$ | hydroxy-oxo-butanal | 0.4 | 1.5 | 0.1 | 1.3 |
| 4 | 85.03 ($C_4H_4O_2$)H$^+$ | butenedial | n.d. | 1.3 | n.d. | 1.2 |
| 3 | 73.03 ($C_3H_4O_2$)H$^+$ | methylglyoxal | 20.9 | 3.2 | 9.5 | 2.7 |
| 3 | 75.04 ($C_3H_6O_2$)H$^+$ | propanoic acid | 14.3 | 1.7 | 13.3 | 1.6 |
| 3 | 89.02 ($C_3H_4O_3$)H$^+$ | pyruvic acid / hydroxy-propanedial | 1.5 | 0.6 | n.d. | 0.6 |
| 2 | 77.03 ($C_2H_4O_3$)H$^+$ | PAN fragment | 3.7 | 0.3 | 3.6 | 0.5 |

*n.d. = not detected

The $C_8$ compounds are dominating both gas and particle phases accounting for 27 % of the total products. The most abundant
$C_8$ gas phase product is m-tolualdehyde ($C_8H_8O$ detected at $m/z$ 121.06), comprising alone 17 % of the reaction products. It is
a first generation ring-retaining aromatic compound previously identified in many studies (Atkinson et al., 1991; Forstner et
al., 1997; Huang et al., 2008; Srivastava et al., 2023; Zhang et al., 2019b; Zhao et al., 2005). Toluic acid is a second generation
ring retaining compound formed by the additional oxidation of m-tolualdehyde by OH radical (Forstner et al., 1997; Srivastava
et al., 2022, 2023), and it is a major $C_8$ particle phase product ($C_8H_8O_2$, detected at $m/z$ 137.06) contributing to 5 % of the SOA
mass. Another major SOA component is found at $m/z$ 171.07 ($C_8H_{10}O_4$) tentatively assigned to dimethyl-hexadienedioic acid,
a ring opening product possibly formed by OH-addition to the benzene ring followed by a ring cleavage, or to dihydroxy-
dimethyl-cyclohexenedione, a ring retaining compound formed by successive OH radical reactions on the benzene ring. A
second ring opening compound is detected at $m/z$ 155.07 ($C_8H_{10}O_3$) previously identified as dimethyl-epoxy-oxo-hexenal
(Zhao et al., 2005). Other $C_8$ compounds are $C_8H_8O_{3-5}$ (at $m/z$ 153.06, $m/z$ 169.05 and $m/z$ 185.05) and $C_8H_{10}O_5$ at $m/z$ 187.06
(Table 2).
The contribution of $C_7$ compounds to gas and particle phases accounts for 6 % and 19 %, respectively. Among the $C_7$ prominent
products in the particle phase we identify $C_7H_8O_2$ at $m/z$ 125.06 tentatively assigned to dimethyl-pyranone (Forstner et al.,
1997) or methyl-hexadienedial resulting from a ring opening of the phenoxy radical intermediate formed via OH-addition to
the ring (Jang and Kamens, 2001). Further oxidation of methyl-hexadienedial will result in methyl-oxo-hexadienoic acid
$C_7H_8O_3$ at $m/z$ 141.05 (Jang and Kamens, 2001). Another $C_7$ compound is the $C_7H_6O_3$ at $m/z$ 139.04 tentatively assigned to
methyl cyclohexene tricarbonyls or hydroxy methyl benzoquinone, both being ring-retaining products. They can be formed
upon OH-addition to first- or second-generation products and release of one methyl group (Jang and Kamens, 2001). Other $C_7$
compounds important in the particle phase include $C_7H_8O$ (detected at $m/z$ 109.06), $C_7H_8O_4$ (detected at $m/z$ 157.05), and
$C_7H_8O_5$ (detected at $m/z$ 173.04). Some of the $C_7$-$C_8$ products are nitrogen containing-compounds: nitro-m-xylene ($C_8H_9NO_2$,
at $m/z$ 152.07), nitrotoluene ($C_7H_7NO_2$ at $m/z$ 138.06), dimethyl nitrophenol ($C_8H_9NO_3$ at $m/z$ 168.06) and nitrocresol
($C_7H_7NO_3$ at $m/z$ 154.05). The latter is the result of $NO_2$ reaction with the benzyl peroxy radical in the H-abstraction route (Li
et al., 2018; Srivastava et al., 2023). Aliphatic nitrogen-containing compounds are easily fragmented in the PTR-MS and thus

are hardly detected. Nevertheless, a known peroxyacetyl nitrate (PAN) fragment $C_2H_4O_3$ detected at $m/z$ 77.03, also previously assigned as an unspecific fragment from nitro-group containing compounds has been detected (Müller et al., 2012). In total, the sum of all nitrogen-containing products (including PAN fragment) accounted for 9-14 % of the gaseous phase mass loading and 6 % of the particulate phase under high-$NO_x$ conditions, which is close to what was seen in toluene photooxidation experiments by Lannuque et al. (2023). Under low $NO_x$ conditions, organic nitrates (including PAN fragment) accounted for 4-12 % in the gas phase and 3-4 % in the particle phase out of total products, indicating limited $NO_x$ influence on the formation of nitrogen-containing compounds (Figs. S1c,d; S2c,d). These results suggest that in our system, the shift from low to high $NO_x$ did not substantially alter the oxidation pathways or enhance $NO_x$-dependent products formation.

The $C_6$ compounds accounted for only 6 % of the gas phase and for 19% of the organic aerosol fraction. Top products are ring-retaining furan-derived, such as methyl-furaldehyde or benzenediols ($C_6H_6O_2$, detected at $m/z$ 111.04) making up 3 % and 6 % in gas and particle phases respectively, and dimethylfurandione ($C_6H_6O_3$, detected at $m/z$ 127.04) with 1.4 % in the gas phase, and 3.3 % in the particle phase, alongside with dimethyl-furanone with 2.2 % in the particle phase ($C_6H_8O_2$, detected at $m/z$ 113.06). Furanoid products can be formed through a bridged oxide intermediate on a bicyclic ring whereas furandiones are known to originate from conjugated dicarbonyls via reaction with OH-radicals followed by cyclization (Forstner et al., 1997; Jang and Kamens, 2001). Examples of unsaturated dicarbonyls and tricarbonyls in the condensed phase are methyl-oxo-pentenal ($C_6H_8O_2$, detected at $m/z$ 113.06), and hydroxy-oxo-hexenal ($C_6H_8O_3$, detected at $m/z$ 129.06), also assigned to methyl-oxo-pentenoic acid (Li et al., 2017, 2018; Zhao et al., 2005). Those multi-carbonyls are a result of the decomposition of bicyclic alkoxy radicals (Huang et al., 2008; Zhao et al., 2005). Other $C_6$ compounds can be found at m/z 143.03 ($C_6H_6O_4$), 97.06 ($C_6H_8O$), 145.05 ($C_6H_8O_4$), and 115.07 ($C_6H_{10}O_2$) (Table 2).

As shown in Fig. 3 and Fig. S1, the $C_5$ products represent a major contributor of the gas phase with around 30 % of the total organic gaseous products and 13 % of the particle phase. In agreement with previous studies, most of the compounds are furan derivatives (Forstner et al., 1997; Jang and Kamens, 2001; Li et al., 2018). The $C_5H_4O_3$ (detected at $m/z$ 113.02) assigned to methylfurandione accounts alone for, 17-19 % of the gas phase products, followed by oxo-pentenal and isomers ($C_5H_6O_2$, detected at $m/z$ 99.04) with 3-7 % abundance. Other important products include unsaturated or oxo-aldehydes (such as $C_5H_6O_2$ at $m/z$ 99.04 and $C_5H_6O_4$ at $m/z$ 131.04), organic acids such oxo-pentenoic acid ($C_5H_6O_3$ at $m/z$ 115.03) (Table 2). Lannuque et al. (2023) identified dihydroxy-oxopentanoic acid (DHOPA, $C_5H_8O_5$), a known tracer of toluene and xylene SOA, in particle phase. In our study, DHOPA parent ion was below the detection limit which is highly probable considering its very low formation yield (Srivastava et al., 2023) and the low precursor concentration used compared to Lannuque et al. (2023). We therefore assume that DHOPA did not contribute significantly to the fragment ion at $m/z$ 131.04 ($C_5H_6O_4$) expected upon water loss. In general the high contribution of $C_5$ compounds can be explained by the m-xylene structure ($C_8$ aromatic) and the presence of two methyl groups on the benzene ring, which implies the formation of methylated furan derivates such as methyl-furandione, methyl-furanone and furaldehyde, in addition to the high contribution of $C_3$ compounds such as methylglyoxal, in agreement with previous studies (Birdsall and Elrod, 2011; Fan and Zhang, 2008; Forstner et al., 1997; Jang and Kamens, 2001; Li et al., 2022; Song et al., 2007; Zhang et al., 2019b).

The $C_4$ compounds are the least abundant in both phases (below 7 %, on average) and account for shorter functionalized
aldehydes, furans and acids formed by minor pathways in the aromatic ring cleavage (Table 2). This is in agreement with the
above explained ring opening mechanism, that mainly breaks the carbon skeleton of the xylene in $C_5$ and $C_3$ compounds,
instead of two $C_4$ moieties (Forstner et al., 1997; Jenkin et al., 2003; Pan and Wang, 2014).
The $C_3$ compounds account for 23-37 % of the gas phase and 6 % of the particle phase products. Methylglyoxal ($C_3H_4O_2$,
detected at $m/z$ 73.03) accounts for 21% of the gas phase products at 295 K while it is close to 10 % at 280 K. It makes up to
3 % of the SOA fraction and is a major second-generation product resulting from the ring cleavage of bicyclic alkoxy radicals
(Fan and Zhang, 2008). Propanoic acid ($C_3H_6O_2$, detected at $m/z$ 75.04) accounts for 8 % of the gaseous phase and only 1 %
in the particle phase, as it is a quite volatile compound. This reaction product can result from the oxidation of multifunctional
ring opening products (Jang and Kamens, 2001).
**3.2.2 Naphthalene**
In analogy with the m-xylene, Fig. 4 presents the chemical distribution of naphthalene gas phase products (Figs. 4a,c) and
particle phase products (Figs. 4b,d) for photooxidation experiments under high $NO_x$ conditions at 280 K, in addition to pie
charts showing the molecular weight distribution. Out of the total naphthalene carbon balance, 30-32 % is accounted for the
detected gas phase products under the different oxidation conditions, taking into account that CO, $CO_2$ and glyoxal are not
measured. As for the SOA formed, it corresponds to 8 % at 295 K and 14 % at 280 K of the carbon balance under high $NO_x$
conditions.

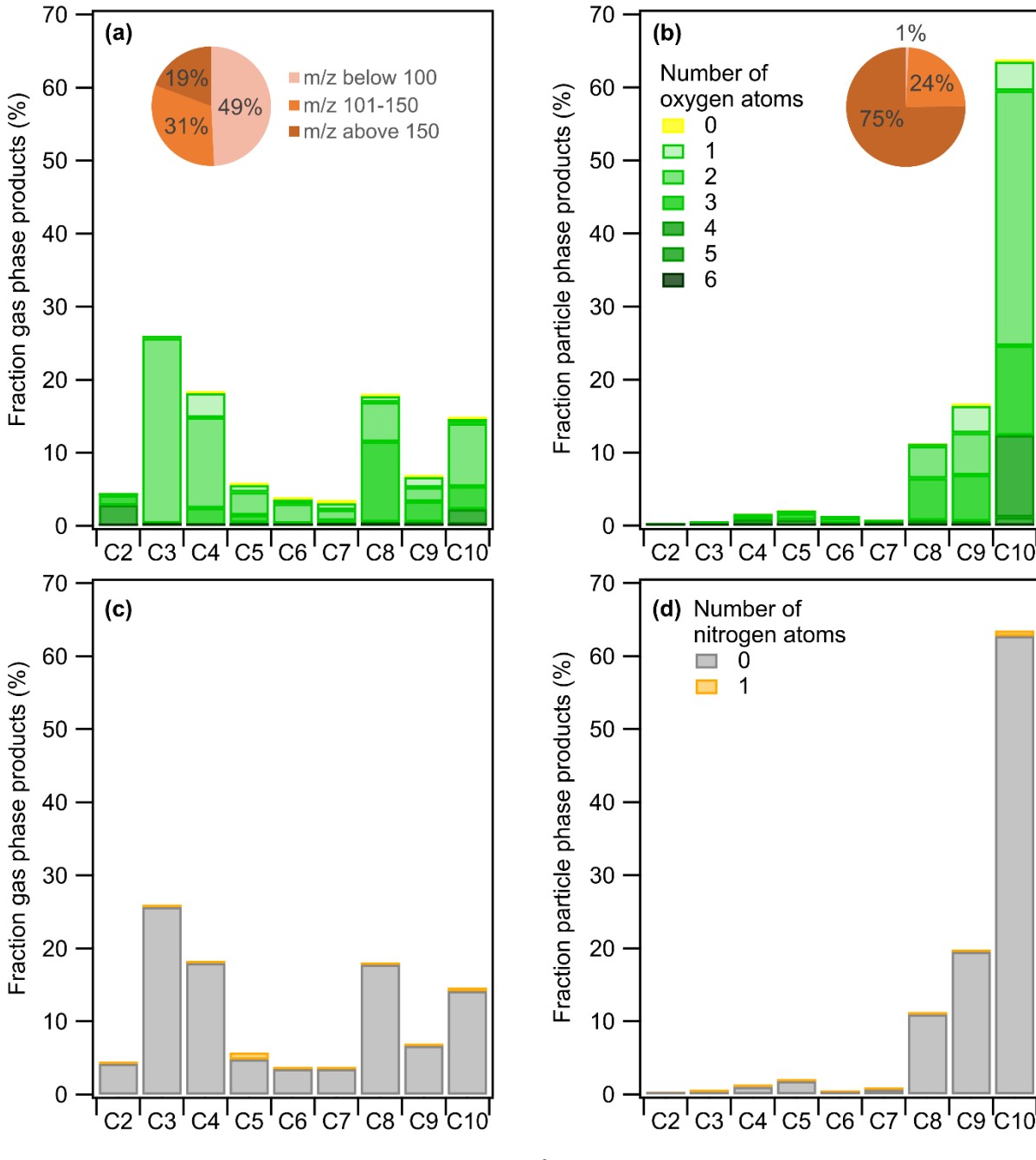

Figure 4. Naphthalene mass products fraction (y-axis) distribution based on the number of carbon atoms (x-axis) for a high $NO_x$ experiment at 280 K, colored by the number of (a,b) oxygen and (c,d) nitrogen atoms. Detected compounds are in the (a,c) gas phase and (b,d) particle phase. Pie charts correspond to the molecular weight contribution to the overall mass.

As the measured carbon distribution is comparable at the two temperature conditions, only the experiment at 280 K is shown in the main text (see Fig. S2 for distributions at 295 K). The gas phase products distribution is dominated by $C_{10}$ and $C_8$ compounds containing mainly 2 or 3 oxygen atoms followed by $C_3$ and $C_4$ compounds, such as methylglyoxal, resulting from the further degradation of naphthalene oxidation products. As expected, the particle phase contains more oxygen atoms and is characterized by heavy molecular weight compounds with $m/z$ above 150 comprising 75 % of the overall mass, explained by the readily partitioning of major $C_{10}$ products containing mainly 2 to 4 oxygen atoms. Table 3 presents the major identified ion fragments, their corresponding chemical formula and a tentative assignment to each compound based on previous studies (Bunce et al., 1997; Chan et al., 2009; Chen et al., 2016; Kautzman et al., 2010; Lee and Lane, 2009; Nishino et al., 2009; Riva, 2013; Sasaki et al., 1997; Tomaz, 2015). Similar to what was observed in the case of m-xylene, the overall oxidation product distributions, including nitrogenous species (1-4 % in both high and low $NO_x$ conditions), remain comparable across $NO_x$ conditions (Figs. S4 and S5 at 295 K). The strong similarity in product distributions and molecular compositions implies minimal shifts in bulk volatility as well in oxidation conditions.

**Table 3. List of ions and their corresponding formulas of the major naphthalene products detected during photooxidation experiments under high $NO_x$. Reaction products are given as a fraction of the gas phase products (in % of µg m$^{-3}$) and as a fraction of the SOA products (in % of µg m$^{-3}$).**

| Carbon number | Measured $m/z$ and ion sum formula | Tentative assignment | $T$ = 295 K | | $T$ = 280 K | |
|---|---|---|---|---|---|---|
| | | | Gaseous products (%) | SOA products (%) | Gaseous products (%) | SOA products (%) |
| 10 | 161.06 $(C_{10}H_8O_2)H^+$ | formyl cinnamaldehyde | 2.0 | 8.9 | 3.5 | 25.4 |
| 10 | 193.05 $(C_{10}H_8O_4)H^+$ | carboxy cinnamic acid | 0.8 | 15.3 | 1.0 | 8.1 |
| 10 | 159.04 $(C_{10}H_6O_2)H^+$ | naphthoquinone | 6.6 | 5.6 | 4.9 | 9.3 |
| 10 | 177.05 $(C_{10}H_8O_3)H^+$ | formyl cinnamic acid | 0.6 | 7.7 | 1.1 | 7.1 |
| 10 | 175.04 $(C_{10}H_6O_3)H^+$ | epoxy-naphthoquinone | 1.6 | 8.4 | 1.9 | 4.8 |
| 10 | 145.07 $(C_{10}H_8O)H^+$ | naphthol | 0.4 | 2.2 | 0.5 | 3.6 |
| 10 | 191.04 $(C_{10}H_6O_4)H^+$ | dihydroxy naphthoquinone | 0.7 | 3.5 | 0.8 | 2.0 |
| 10 | 195.06 $(C_{10}H_{10}O_4)H^+$ | carboxybenzenepropanoic acid benzofurancarboxyaldehyde | 0.3 | 1.5 | 0.3 | 1.0 |
| 10 | 174.05 $(C_{10}H_7NO_2)H^+$ | *nitronaphthalene* | 1.2 | 0.3 | 0.3 | 0.2 |
| 10 | 190.05 $(C_{10}H_7NO_3)H^+$ | *nitronaphthol* | 0.5 | 2.5 | 0.1 | 0.4 |
| 9 | 147.05 $(C_9H_6O_2)H^+$ | benzopyrone | 3.5 | 8.3 | 1.9 | 5.8 |
| 9 | 163.04 $(C_9H_6O_3)H^+$ | hydroxycoumarin | 1.5 | 6.3 | 2.2 | 4.5 |
| 9 | 133.06 $(C_9H_8O)H^+$ | indanone | 1.6 | 1.3 | 1.3 | 3.6 |
| 9 | 179.04 $(C_9H_6O_4)H^+$ | dihydroxy-indanedione | 0.5 | 4.2 | 0.1 | 2.8 |
| 9 | 165.06 $(C_9H_8O_3)H^+$ | hydroxy cinnamic acid | 0.9 | 2.7 | 0.6 | 1.8 |
| 9 | 181.05 $(C_9H_8O_4)H^+$ | dihydroxy cinnamic acid | 0.2 | 1.5 | 0.2 | 0.9 |
| 8 | 135.05 $(C_8H_6O_2)H^+$ | phthaldialdehyde / phthalide | 19.3 | 3.9 | 5.4 | 4.3 |
| 8 | 149.03 $(C_8H_4O_3)H^+$ | phthalic anhydride | 13.8 | 3.3 | 9.9 | 2.0 |

| | | | | | | |
|---|---|---|---|---|---|---|
| 8 | 151.04 (C$_8$H$_6$O$_3$)H$^+$ | hydroxy phthaldehyde hydroxy phthalide | 0.5 | 1.2 | 1.0 | 3.7 |
| 7 | 123.05 (C$_7$H$_6$O)H$^+$ | benzoic acid | 1.9 | 0.2 | 1.4 | 0.1 |
| 7 | 107.05 (C$_7$H$_6$O)H$^+$ | benzaldehyde | 1.0 | n.d. | 0.9 | n.d. |
| 6 | 111.05 (C$_6$H$_6$O$_2$)H$^+$ | catechol / benzoquinone | 1.2 | n.d. | 0.9 | n.d. |
| 6 | 115.07 (C$_6$H$_{10}$O$_2$)H$^+$ | dimethyloxolan-one / hexanedione | 1.3 | 0.3 | 1.3 | 0.6 |
| 5 | 101.06 (C$_5$H$_8$O$_2$)H$^+$ | methyl furan | 0.7 | n.d. | 1.6 | n.d. |
| 4 | 89.06 (C$_4$H$_8$O$_2$)H$^+$ | hydroxybutanone / hydroxybutanal | 10.1 | 0.1 | 11.5 | n.d. |
| 4 | 71.05 (C$_4$H$_6$O)H$^+$ | dihydrofuran / MACR / MVK | 1.2 | n.d. | 1.8 | n.d. |
| 3 | 75.04 (C$_3$H$_6$O$_2$)H$^+$ | propanoic acid | 12.3 | 0.3 | 19.9 | 0.2 |
| 3 | 73.03 (C$_3$H$_4$O$_2$)H$^+$ | methylglyoxal | 3.6 | 0.1 | 5.4 | 0.1 |
| 2 | 77.02 (C$_2$H$_4$O$_3$)H$^+$ | PAN fragment | 2.0 | 0.1 | 1.3 | 0.1 |

$^*$n.d. = not detected
The C$_{10}$ compounds account for around 15 % of the gas phase products at both temperatures and clearly dominate the particulate
phase with 59 % and 64 % of the overall aerosol mass at 295 K and 280 K, respectively. The 2-formyl cinnamaldehyde
(C$_{10}$H$_8$O$_2$ at $m/z$ 161.06) is the most abundant particle phase product accounts for 25 % of the SOA formed at 280 K. It is
assumed to be formed via two possible routes i) ring cleavage upon the reaction of the naphthol peroxy radical (from an OH-
naphthalene adduct) with NO, and ii) a hydrogen shift from the alcohol group on the naphthol peroxy radical followed by loss
of OH and ring opening (Kautzman et al., 2010; Nishino et al., 2009; Qu et al., 2006; Sasaki et al., 1997; Wang et al., 2007).
The predominance of one route over the other is depending on the amount of NO$_x$. The 2-formyl cinnamaldehyde can further
be oxidized, leading to the formation of 2-carboxy cinnamic acid (C$_{10}$H$_8$O$_4$ at m/z 193.05), the second most important C$_{10}$
compound that accounts for 8-15 % of the SOA mass. The oxidation of 2-formyl cinnamaldehyde also produces 2-formyl
cinnamic acid (C$_{10}$H$_8$O$_3$ at $m/z$ 177.05) (Bunce et al., 1997), that accounts for 7-8 % of the condensed phase. The addition of
O$_2$ to the OH-naphthalene adduct results in the formation of epoxy-quinone (C$_{10}$H$_6$O$_3$ at $m/z$ 175.04), representing 5-8 % of
the particle phase. Other important C$_{10}$ ring retaining compounds are dihydroxy naphthoquinone (C$_{10}$H$_6$O$_4$ at $m/z$ 191.04) and
1,4-naphthoquinone (C$_{10}$H$_6$O$_2$ at m/z 159.04). This latter has multiple formation routes, either from reaction of OH radical
with naphthalene or with naphthol (C$_{10}$H$_8$O at $m/z$ 145.07), or from the photodegradation of nitronaphthalene (C$_{10}$H$_7$NO$_2$ at
$m/z$ 174.05) (Atkinson et al., 1989; Kautzman et al., 2010), potentially explaining the low abundance of the latter (< 1 % in
mass) alongside with its reaction product with OH, nitro-naphthol (C$_{10}$H$_7$NO$_3$ at $m/z$ 190.05). At both high and low NO$_x$
conditions, the NO$_2^+$ signal in the particle phase did not exceed 3 % out of the aerosol mass detected by CHARON-PTR-ToF-
MS meaning that the nitro-derivates of naphthalene did not undergo strong fragmentation. In total, the sum of all nitrogen-
containing products (including PAN fragment) accounted for 2-3 % of the gaseous phase mass loading and 1-4 % of the
particulate phase under high-NO$_x$ conditions. The observed low yields of nitro derivatives during the photo oxidation of
naphthalene is in agreement with previous studies (Kautzman et al., 2010; Lee and Lane, 2009; Sasaki et al., 1997).
Nitronaphthol (C$_{10}$H$_7$NO$_3$ at m/z 190.05) is a well known tracer of naphthalene SOA in high NO$_x$ conditions, and has been

observed here as a main nitrogen containing compound. Other previously reported tracers (nitrosalicylic acid, dinitrosalicylic acid or nitrophthalic acid) are not detected by CHARON-PTR-ToF-MS. The $NO_x$ conditions in our study might explain they are not formed, since the formation of these compounds is $NO_x$ dependent, and our study used lower $NO_x$ levels compared to studies using HONO or $CH_3NO$ as OH radical precursors, for example (Sato et al., 2022). It should also be considered that the UV lights used, which peaked at 310 nm, may induce photolysis of nitronaphthalene and other nitro-derivates, as previously reported in chamber experiments with similar UV lamps (Healy et al., 2012).

The formation of $C_9$ and $C_7$ products can be explained by H-abstraction of 2-formyl cinnamaldehyde, leading to a formyl peroxy radical that subsequently reacts with NO to form an alkoxy radical implying the loss of $CO_2$ group (Kautzman et al., 2010). The $C_9$ compounds contribute by 6-9 % to the gas phase and are the second highest contributors to the particle phase with 16-20 % of the total organic mass. Identified chemical formulas include $C_9H_6O_2$ (at $m/z$ 147.05) which can be either benzopyrone (also known as coumarin) or indene-dione, as well as indanone ($C_9H_8O$ at $m/z$ 133.06) first detected by Lee and Lane (2009). No formation mechanism has been proposed for these products so far, but their molecular structure clearly indicates a rearrangement following ring opening, to form either a new ring at 6 atoms including one oxygen in the case of coumarin, or at 5 atoms for the indene-dione and indanone. This ring closure is a common feature in the case of dialdehydes oxidation, as explained by Lannuque and Sartelet (2024), which may support the hypothesis that these compounds originated from 2-formyl cinnamaldehyde. Further photo-oxidation of coumarin may lead to hydroxycoumarin ($C_9H_6O_3$ at $m/z$ 163.04) accounting for 2.2 % of the gas phase and 4-6 % of the particle phase. A more oxygenated compound, dihydroxy-indanedione ($C_9H_6O_4$ at $m/z$ 179.04), was previously identified by Lee et al. (2018). Logically, this compound was mainly present in the condensed phase with 3-4 % of the total SOA mass.

$C_8$ oxidation products make up a good 18-35 % and 10-11 % of the gas and particle phases, respectively. Phthaldialdehyde and phthalic anhydride largely dominate the $C_8$ compounds and are of particular importance in the gas phase. Phthaldialdehyde ($C_8H_6O_2$ at $m/z$ 135.05), is both a first- and second-generation ring-opening product of the OH radical reaction with naphthalene, with the second-generation pathway expected to originate from the reaction of OH radical with 2-formyl cinnamaldehyde (Sasaki et al., 1997; Wang et al., 2007). Further addition of OH radical to phthaldialdehyde will lead to hydroxy phthaldehyde / hydroxy phthalide ($C_8H_6O_3$ at $m/z$ 151.04) that makes up 3.7 % of SOA (Table 3), while the H-abstraction followed by intramolecular cyclization results in phthalic anhydride ($C_8H_4O_3$ at m/z 149.03) making up 2-3% of the condensed phase, and 10-14 % in gas phase (Table 3) previously observed by Wang et al. (2007).

$C_7$ compounds are relatively less abundant accounting for 3 % of the gas phase products and less than 1 % of SOA. They include compounds like benzoic acid ($C_7H_6O_2$ at $m/z$ 123.05) and benzaldehyde ($C_7H_6O$ at $m/z$ 107.05).

Only two $C_6$ compounds are detected and represent 3-4 % of the total gas products and only 1 % of the SOA yield. The dominant $C_6$ in the gaseous phase is $C_6H_{10}O_2$ (at $m/z$ 115.07) with 1.3 %, which can be dimethyloxolanone or hexanedione. The former can be formed from the oxidation of phthaldialdehyde, implying the opening of the second aromatic cycle (Kautzman et al., 2010). Another minor $C_6$ compound is catechol or benzoquinone ($C_6H_{10}O_2$ at $m/z$ 115.07).

$C_5$ products account for only 3-6 % of the gaseous phase and less than 2 % of SOA. A major detected compound is $C_5H_8O_2$ at
$m/z$ 101.06 (1.6 %) which can be assigned as methyl furan or 4-oxopentanal.
The $C_2$-$C_4$ products are more volatile, all together make less than 2 % of the total SOA mass but present a considerable fraction
of the gas phase products with 18 % for $C_4$, 26 % for $C_3$, and 4 % for $C_2$. The major $C_4$ contributor is $C_4H_8O_2$ (at $m/z$ 89.06)
representing 12 % of the gas phase products. It is tentatively assigned to hydroxybutanal or butanoic acid. Almost all the $C_3$
fraction is represented by propanoic acid ($C_3H_6O_2$ at $m/z$ 75.04) with 12-20 % contribution to the gas phase products, and
methylglyoxal ($C_3H_4O_2$ at $m/z$ 73.03) up to 5 %. The $C_2$ compound is tentatively assigned to the PAN fragment $C_2H_4O_3$ detected
at $m/z$ 77.02 (1.4 %) (Müller et al., 2012).
**3.3 Experimentally derived and estimated gas-particle phase partitioning**
A two-dimensional space, the 2D-VBS framework (Donahue et al., 2011; Murphy et al., 2012), is used as a means to visualize
the compounds distribution as function of the experimentally derived volatility ($\log_{10}C_i^*$) and the O/C ratio (Fig. 5) or the
oxidation state of carbon (OSc) (Fig. S4). Such molecular level effect of temperature on SOA for m-xylene and naphthalene
is reported here for the first time. Figures 5a and 5b display the distribution of measured volatilities for the major compounds
detected in both gas and particle phases for m-xylene and naphthalene, respectively, under high $NO_x$ conditions. The volatilities
of the identified compounds are in the range of SVOCs with $C_i^*$ values ranging from 0.3 to 300 μg m$^{-3}$ (vapor pressures
approximately $10^{-8}$ to $10^{-5}$ torr) to IVOCs with $C_i^*$ values ranging from 300 to $3 \times 10^6$ μg m$^{-3}$ (vapor pressures approximately
$10^{-5}$ to $10^{-1}$ torr) (Donahue et al., 2012), indicated by the light green ($\log_{10}C_i^*$ from 0 to 2.5) and light blue ($\log_{10}C_i^*$ from 2.5
to 5) background shading, respectively. Data points are indicated for the experiments at 280 K (purple circles) and 295 K (red
circles), and the size of the circles is proportional to the mass concentration (in μg m$^{-3}$) of each ion in the particle phase.
For m-xylene approximately out of 110 ions detected in the particle phase, half of them are exclusively found in the particle
phase. This fraction should populate a low volatility area (as ELVOC), and it is not represented in this 2D-VBS (Fig.5a) but
represents more than 30 % of the SOA mass. Among the remaining ions that partition between the two phases, only 18 are
considered in Fig.5a, either because they are important in terms of mass or because they are considered parent ions following
Gkatzelis et al. (2018) method. About 24 to 58 % of the particle mass populates the SVOCs regime, while 8 to 16 % are in the
IVOCs regime, depending on the experimental conditions. In a recent work the SOA components from OH radical oxidation
of toluene populated mostly the SVOC range and only 10-17 % was exclusively in the particle phase (Lannuque et al., 2023),
probably because of higher initial VOC concentration and lower OH exposure (10-20 h compared to 1.3-3 days in this study)
leading to a less advanced oxidation. But overall, the effect of temperature observed is quite similar for the two chemical
systems. For naphthalene at 280 K, out of the 110 ions detected in the particle phase, 40 ions are exclusively observed in the
particle phase, making up 5 % only of the condensed mass fraction. For the selected 20 compounds that partition between the
two phases, approximately 78-95 % of their SOA mass populates the SVOC regime and 3-17 % lies in the IVOC portion. The
larger skeleton structure of naphthalene oxidation products, even first generation, can explain the high fraction of SOA in the
SVOC regime (predominantly $C_{8-10}$ compounds) as confirmed by the O/C range, lower for naphthalene than for m-xylene.
The derived saturation concentration ($C_i^*$) values range between 1 to 6919 μg m$^{-3}$, comparable to other SOA systems (Gkatzelis
et al., 2018; Kostenidou et al., 2024; Lannuque et al., 2023). Previous studies on biogenic VOCs reported a decreasing volatility
with increasing OSc (Gkatzelis et al., 2018; Jimenez et al., 2009; Kroll et al., 2011), while the present works and previous
investigations on anthropogenic VOCs (toluene and gasoline vehicle emissions) did not confirm such a trend (Kostenidou et
al., 2024; Lannuque et al., 2023).
Figures 5c and 5d present the difference of $\log_{10}C_i^*$ between the two experiments conducted at 280 and 295 K, $\Delta\log_{10}C_i^*$. Its
values vary from 0.06 to 1.08 as a function of carbon number and oxygen number (tables S2 and S3). For m-xylene oxidation
products (Fig. 5c), $\Delta\log_{10}C_i^*$ decreases with increasing carbon number. Indeed, the C$_5$-C$_8$ products bearing 3 to 5 oxygen atoms
(darker green markers) lie at the bottom of the plot generally exhibiting $\Delta\log_{10}C_i^*$ values below 0.4, emphasizing the effect of
oxygen atoms in reducing the volatility of these compounds. The C$_3$ compounds, tentatively associated to methylglyoxal and
propanoic acid, exhibit also moderate $\Delta\log_{10}C_i^*$, probably because they are still volatile even at the lower experimental
temperatures (Table 2). Only few among the identified C$_4$-C$_5$ products exhibit $\Delta\log_{10}C_i^*$ values above 0.6 and are tentatively
identified as ring opening products, with multiple functional groups increasing their polarity (aldehydes, acids and
furandiones).

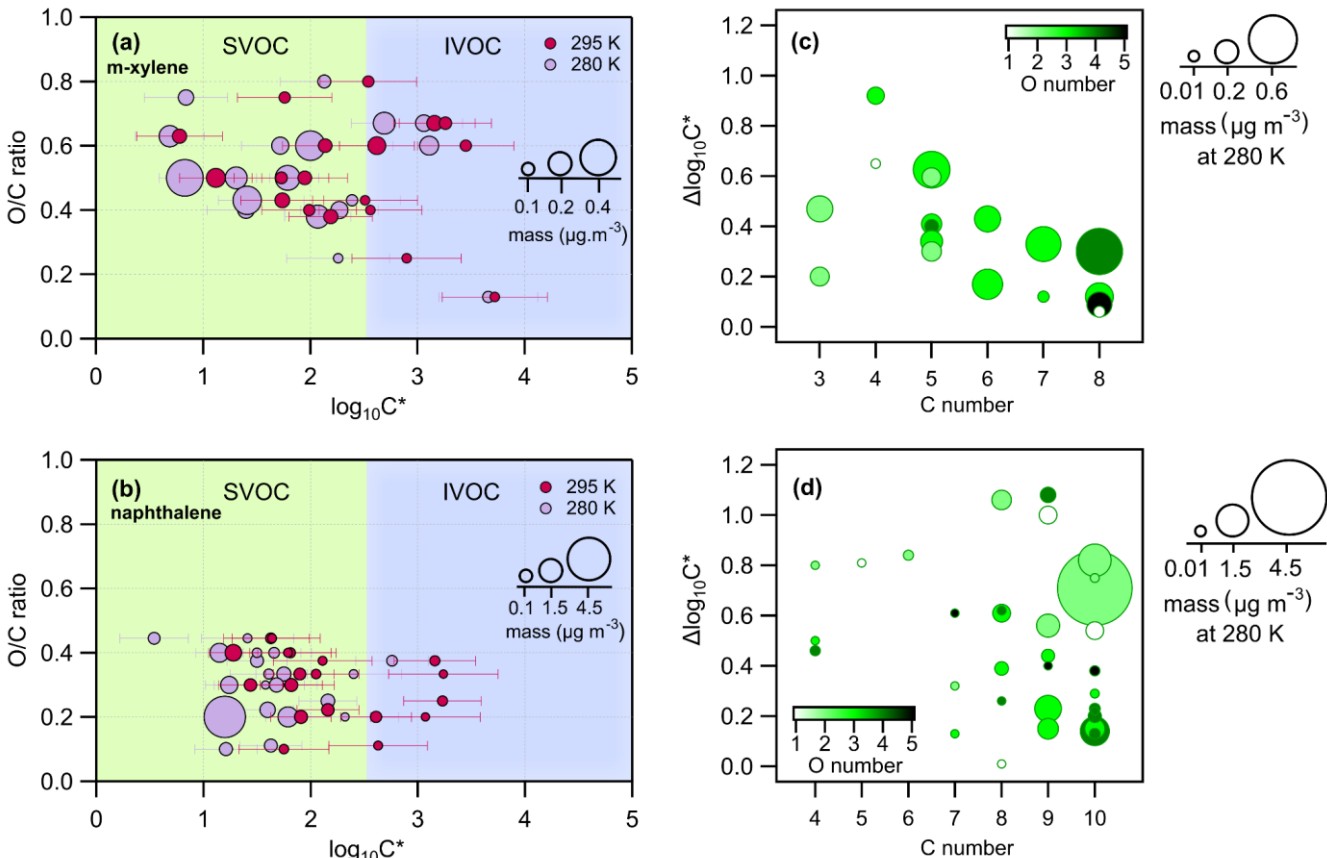

**Figure 5. Panels (a) and (b) present the 2D-VBS framework: the O/C ratio (y-axis) for the detected parent ions of m-xylene and**
**naphthalene photooxidation, respectively, as a function of saturation concentration ($\log_{10}C_i^*$ in µg m$^{-3}$; x-axis) for high NO$_x$**
**experiments. The size of the circles denotes the mass of each species. Experiments carried out at 280 K are in light violet while the**
**ones carried out at 295 K are in magenta. The light green and the light blue background shadings correspond to the SVOC (0 <**
**$\log_{10}C_i^*$ < 2.5) and the IVOC (2.5 < $\log_{10}C_i^*$ < 5) regimes, respectively. Panels (c) and (d) present $\Delta\log_{10}C_i^*$ (the difference of $\log_{10}C_i^*$**
**between experiments at 280 K and 295 K; y-axis) as function of carbon number (x-axis), color scaled by oxygen number and sized**
**by the mass (in µg m$^{-3}$) of each compound at 280 K for m-xylene and naphthalene oxidation products, respectively.**

Naphthalene oxidation products on the other hand (Fig. 5d) have $\Delta\log_{10}C_i^*$ over a similar range compared to m-xylene, but
exhibit a different behaviour. Most of the relevant reaction products are associated to $C_{8-10}H_{4-10}O_{1-4}$ products, accounting for
90 % of the SOA mass and have at least one aromatic ring. These compounds belong SVOC with a $\log_{10}C_i^*$ < 2, and span a
broad range of $\Delta\log_{10}C_i^*$ depending on the number of oxygen atoms (from 1 to 5) and the specific functionalities. Logically,
the most oxygenated $C_{10}$ are poorly affected by temperature, since they are the least volatile and already mostly in the particle
phase at 295 K. Compared to m-xylene, the most important naphthalene SOA-products are less oxygenated and seem to be
more temperature-sensitive.

## 3.4 Comparison of experimentally derived and calculated volatilities

The experimentally derived volatilities are here compared to the estimated ones from pure-liquid saturation vapour pressure using Volcalc based on SIMPOL.1 (Meredith et al., 2023; Pankow and Asher, 2008; Riemer, 2023). Volatilities from the two methods are presented in Figs. 6a and 6b for both m-xylene and naphthalene oxidation products at 280 K, respectively. A considerable discrepancy is observed between the experimental and calculated values, the latter ones spanning a larger range of volatilities. For m-xylene SOA, the theoretical approach tends to largely overestimate the volatility of small and oxygenated compounds below $m/z$ 120, similarly to recent investigations using different techniques as SV-TAG coupled to GC-MS, thermal desorption-AMS and FIGAERO-CIMS and CHARON (Stark et al., 2017; Ijaz et al., 2024; Liang et al., 2023).

For some light carbonyl compounds present among the m-xylene reaction products, the disagreement can be potentially explained by presence of ammonium ions ($NH_4^+$) in the seeds that can act as a catalyst for accretion reactions such as aldol condensation (Li et al., 2022, 2011; Nozière et al., 2010; Sareen et al., 2010) as well as acetal and hemiacetal formation (Jang et al., 2002; Li et al., 2022; Loeffler et al., 2006; Shapiro et al., 2009) shifting the equilibrium to the condensed phase for these molecules. Furthermore, Lannuque et al. (2023) have achieved better model representation of experimental SOA mass concentration after including interactions between aldehydes as methylglyoxal and inorganic compounds (such as ammonium). Those reactions are also catalyzed in the presence of water, especially for highly oxidized hydrophilic compounds (Meng et al., 2024). For some other compounds, as functionalized acids and some dialdehydes, disagreements may arise from the fragmentation in the mass spectrometer. Despite a relatively low $E/N$ value (68 Td) used in this work, fragmentation may still occur, particularly in polyfunctional products (Leglise et al., 2019). We also tested the fragmentation of several compounds in a separate work (Lannuque et al., 2023) where we could observe low or negligible fragmentation for methylglyoxal, furans, furfurals, maleic acids and anhydrides but important fragmentation for small linear aldehydes.

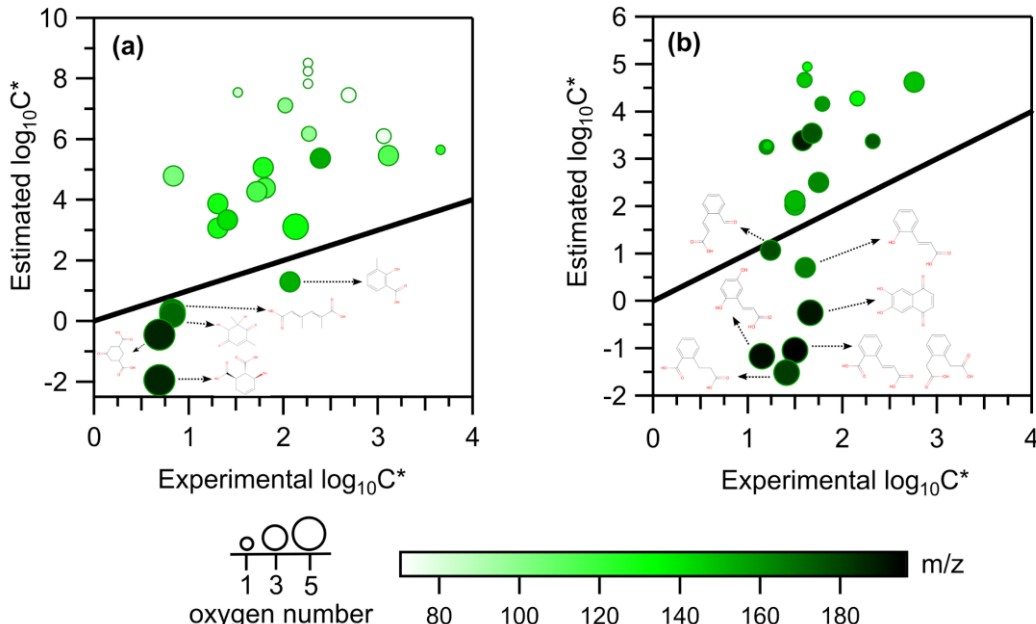

**Figure 6. Theoretical (y-axis) versus experimental $\log_{10}C_i^*$ values (x-axis) for (a) m-xylene and (b) naphthalene oxidation products at 280 K under high NOx conditions. The black line is the 1:1 fit. The size of the markers is proportional to the oxygen number. The color gradient corresponds to *m/z* of the detected compounds.**

For naphthalene SOA products, larger discrepancies are observed for larger compounds ($m/z \geq 160$) associated to $C_{8-10}$ compounds bearing multiple functional groups. For such types of compounds, the model predicts very low volatilities (O'Meara et al., 2014) also demonstrating that SIMPOL.1 consistently overestimates temperature-induced volatility changes for highly oxygenated compounds (such as hydroxylated ketones and diacids). These discrepancies are attributed to the additive nature of SIMPOL.1's functional group contribution framework. At 280 K, a carboxylic acid group (-COOH) decreases the saturation vapor pressure by nearly a factor of 6000, a hydroxyl group (-OH) by factor 200, while a ketone group (=O) reduces it by less than a factor of 9. It is also worth noting that multiple structural isomers are not distinguished in our work, and this may also introduce some uncertainties. An example is provided by 1,2-naphthoquinone and 1,4-naphthoquinone owing quite different vapor pressure making the former potentially more interesting in the particle phase (McWhinney et al., 2013). Liang et al. (2023) and Dang et al., (2019) also reported very different gas-particle partitioning behavior and vapor pressures values among isomers. Barley and McFiggans (2010) and Peräkylä et al. (2020) highlighted significant overpredictions for diacids and polyfunctional compounds, attributing these errors to the model's parameterizations and group contributions. This work therefore aligns with previous findings and suggests that vapor pressures of multifunctional compounds of the aerosol are still not well estimated by models.

Figure S7 shows the comparison of estimated versus experimental $\log_{10}C_i^*$ values at 295 K under high NOx conditions for both m-xylene and naphthalene. These results closely match those at 280 K (Fig. 6), with no significant difference in the distribution.

Figure S8 presents the same analysis under low $NO_x$ conditions at 295 K, which similarly shows no notable deviation from the
trends observed at low $NO_x$ (Fig. S7). This indicates that neither temperature nor $NO_x$ regime substantially explain the model-
observation discrepancy in our dataset. Temperature-induced volatility changes are quantified in Figs. 5c,d which represents
the difference between experimental $\log_{10}C_i^*$ at 280 K and 295 K (i.e. $\Delta\log_{10}C_i^*$) against the C number. The temperature effect
on the discrepancy between the experimental and the estimated $\log_{10}C_i^*$ values is presented in Fig. S9. The values of $\Delta\log_{10}C_i^*$,
the difference between estimated and experimental $\log_{10}C_i^*$, for the oxidation products of m-xylene (Fig. S9a) and naphthalene
(Fig. S9b) is plotted as a function $m/z$, at 295 K and 280 K. The discrepancies at both temperatures are quite close to each
other. Overall, SIMPOL.1's bias in predicting volatility for oxidation products is not strongly affected by temperature. This
consistent discrepancy with temperature and $NO_x$ confirmed that partitioning of organic compounds is more complex than the
simple evaporation of compounds isolated, and that surface interactions and bulk chemical reactions are potential important
drivers affecting partitioning equilibria.
**4 Conclusion**
This study presents a detailed experimental investigation of the OH-initiated photooxidation of two anthropogenic aromatic
precursors, m-xylene and naphthalene, using an OFR under different $NO_x$ and temperature conditions. For both precursors,
SOA yields are found to strongly increase when temperature and $NO_x$ decrease, in agreement with previous studies. The
CHARON-inlet coupled to a PTR-ToF-MS successfully quantified between 65-80 % of the total organic mass covered by an
aerosol mass spectrometer (HR-ToF-AMS). Major products both in the gas and particulate phases are confirmed based PTR-
ToF-MS inferred molecular formula and intercomparison with the literature. Major gas phase products of m-xylene SOA are
$C_3$, $C_5$ and $C_8$ compounds whereas the particle phase products are dominated by $C_6$-$C_8$ compounds. This pattern is consistent
through different temperatures, indicating similar chemistry in these conditions. For naphthalene experiments, gas phase
products are dominated by $C_8$ and $C_{10}$ compounds, while the particle phase composition mainly consists of $C_8$, $C_9$ and $C_{10}$
compounds. Nitro-derivatives (nitrogen containing compounds + PAN fragment) measured in both phases did not exceed 7 %
in naphthalene experiments whereas they could reach up to 20 % in the m-xylene experiments. The similarity in product
distributions under both high and low $NO_x$ conditions for both m-xylene and naphthalene indicates that the chemical reactions
were largely unaffected by the $NO_x$ levels, implying that $NO_x$ had minimal influence on oxidant concentrations in the OFR.
The volatility properties of the individual compounds are presented in the 2D-VBS framework: 24-58 % of the SOA mass
generated by m-xylene populates the SVOC regime and 8-16 % populates the IVOCs regime, while the naphthalene SOA
mass is mostly (up to 95 %) found in the SVOC regime. No clear correlation could be observed between the volatility values
and the increasing oxidation state (OSc), in agreement with previous studies on anthropogenic monoaromatic precursors.
Temperature variation, from 295 to 280 K, induced an expected decrease in volatility ($\Delta\log_{10}C_i^*$), this decrease ranging from
0.06 to 1.08. The magnitude of $\Delta\log_{10}C_i^*$ seems to be controlled by multiple parameters, as temperature, carbon number, oxygen
number and specific chemical moieties. When experimentally derived volatilities are compared to a group contribution

parameterization model based on pure liquid vapor pressure (SIMPOL.1), large discrepancies are observed. The small discrepancies between estimated and experimental $\log_{10}C_i^*$ at both temperatures indicate that the temperature has little impact on SIMPOL.1's bias in predicting the volatility of oxidation products. The difference between estimated and experimental volatilities highlight the complexity of gas-particle partitioning, and the limit of parameterizations that should be further validated using larger datasets from various measurement techniques and experimental conditions.

This study advances our understanding of SOA composition and gas-particle partitioning at different and relevant atmospheric conditions and also holds some implications for urban air quality management and climate modelling. The observed temperature-dependent shifts in SOA mass loadings and volatility highlight the importance of accounting for seasonal variations, particularly in urban areas with high anthropogenic VOC emissions. Furthermore, the findings underscore the need to refine current air quality and climate models by incorporating more recent findings and real-world atmospheric conditions, paving the way for more accurate predictions of aerosol impacts on air quality, climate, and public health.

**Data availability**

Finalized data are mostly available in supplementary material. More detailed data are available on request.

**Supplementary material**

**Competing interests**

The authors declare that they have no conflict of interest.

**Author contribution**

MS, JK, and BD designed the experimental setup. MS, JK, BTR and BD performed the experiments and data treatment. MS, BD and JK analysed and interpreted the data. MS ran the SIMPOL.1 model. MS drafted the article. All the co-authors revised the article.

**Financial Support**

This work was funded by the POLEMICS project of the Agence Nationale de la Recherche (ANR) program (grant ANR-18-CE22-0011), and the MAESTRO-EU6 project (ADEME CORTEA n. 1866C0001), the French government under the France 2030 investment plan, as part of the Initiative d'Excellence d'Aix-Marseille Université – A*MIDEX " AMX-21-PEP-016, the French national research agency (ANR-22-CE22-0003-01) and the Environmental Sciences doctoral school (ED 251). BD

acknowledges the MITI program from CNRS for the financial support allowing the design and the construction of the new
oxidation flow reactor (OFR) deployed in this study.

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
