# Peer review of "Gas-particle partitioning of m-xylene and naphthalene oxidation"

_EGUsphere, 2025_

## Author Response (AR1)

**We thank the reviewers for their valuable and constructive comments. In this document, the reviewers' comments are presented in black, while authors' responses are provided in blue. Any text that has been added to or modified in the manuscript is presented in blue italics and enclosed in quotation marks.**

**#reviewer number 1 (RC1)**

Reviewers Comments: The authors present an evaluation of partitioning behavior of individual SOA products from H-initiated photooxidation of two anthropogenic aromatic precursors: m-xylene and naphthalene as well as atmospheric condition effects on the portioning such as temperature and NOx/VOC ratio. M-xylene and naphthalene are used as initial VOC and then react with OH radicals, and thus, to quantify the gas-particle partitioning using proton transfer reaction time-of-flight mass spectrometer (PTR-ToF-MS) coupled to a CHemical Analysis of aeRosol ONline (CHARON). The results show, in the case of oxidation of m-xylene, that gas-phase major products are C3, C5 and C8 whereas particle-product distributions exhibit a progressive increase from C2 to C8. In contrast, naphthalene products partition more readily into the condensed phase, with C8-C10 products dominating. Also, low temperature has enhanced yield of SOA formation, but low VOC/Nox ratios reduce it. Several oxidation products from both of m-xylene and naphthalene exhibits a volatility distribution in the SVOC regime, with fewer in the IVOC regime. A poor disagreement between model prediction and measurements is observed using SIMPOL. This work provides important insights into SOA formation processes in oxidation of m-xylene and naphthalene, with implications for improving understanding of gas-particle partitioning. To be summary, the manuscript is well written. However, the most important issue is lacking of explanation in the manuscript. Comparing with the previous studies, there is consistent findings between this and earlier studies.

General and specific comments are listed below:

1.      Abstract and Introduction:

Authors start the introduction of the manuscript with importance of xylene and naphthalene. A lot of references are cited to introduce effect of atmospheric condition as temperature and NOx/VOC ratios. From the current introduction, it is not detailed introduction for why this study is clearly important to represent for atmospheric SOA formation at low temperature. In conclusion, the introduction requires an improvement.

We thank the reviewer for highlighting the need to more clearly articulate the motivation for this study. In response, we have revised the introduction to explicitly emphasize the importance of investigating the temperature dependence of SOA formation at the molecular level. While temperature is widely acknowledged as a key factor influencing gas-particle partitioning, most previous studies have focused on overall SOA yields or employed modeled volatility distributions. As such, they often lack molecular-level resolution, particularly for oxidation products of aromatic VOCs. We now clarify that experimental data examining the temperature effect on individual product partitioning remains scarce, and that such measurements are essential for improving volatility parameterizations in atmospheric models under colder conditions, such as those prevalent during nighttime or winter.

The following paragraph is added in the *Introduction, Lines 85-92* of the revised manuscript:

*"Although temperature is recognised as a major factor governing gas-particle partitioning, most prior studies have primarily focused on the bulk SOA yields or selected classes of compounds (Zhang et al., 2023; Takekawa et al., 2003; Svendby et al., 2008; Ahn et al., 2021; Rutter et al., 2007; John et al., 2018; Bahrami et al., 2024; Zhou et al., 2021; Wei et al., 2016). While these approaches have advanced the understanding of SOA formation and representation in models, they often lack molecular level resolved resolution. Experimental studies that probe the effect of temperature on the partitioning of individual SOA oxidation products remain limited and relatively recent (Deng et al., 2021; Li et al., 2024; Lannuque et al., 2023; Fan et al., 2025). Such measurements are particularly important for refining SOA volatility parameterizations under atmospherically relevant cold conditions, such as those occurring during nighttime or wintertime episodes."*

3. Results and discussion

3.1 SOA yield formation: author shows that measured SOA-derived naphthalene mass concentration is consistent with that of previous results with different temperature under environmental conditions. However, in the case of xylene oxidation at high and low temperature, an inconsistent trend between the previous studies and this study. Authors are supposed to explain this inconsistence.

We would like to clarify that the temperature-dependent SOA yield trends observed for m-xylene in our study are consistent with previous literature (see Figure 2). Specifically, under low-$NO_x$ conditions, our results closely agree with those of Takekawa et al. (2003), who observed a similar doubling of yield when decreasing the temperature from 303 K to 283 K. This is also in line with the findings of Lannuque et al. (2023), who reported high SOA yields at lower temperatures (for toluene).

3.2 Chemical composition of oxidation products in gas and particle phases

The chemical composition has been described in the whole section. Author is recommended to add some discussion on chemical composition under the different environmental conditions.

As expected, higher contribution of organonitrate products has been observed for high $NO_x$ conditions, but the fraction of organonitrates is moderate in all conditions. To support this discussion, we updated Figs. 3, 4, S1, and S4 (Fig. S4 previously denoted as Fig. S3) to identify the contribution of N-compounds, and added new Figs. S2 and S5 in the Supplement (also presented below). We have also expanded our discussion to include $NO_x$ effects on chemical composition in the revised manuscript: *(Section 3.2.1, Lines 334-341, 380-386; Section 3.2.2, Lines 443-446, 468-478).*

[revised manuscript text omitted]

3.3 Experimentally derived and estimated gas-particle phase partitioning

Using 2D-VBS framework to calculate volatility properties of the individual compounds. The description of 2D-VBS framework is needed in the experiment section.

We would like to clarify that the $C_i^*$ values plotted in the 2D-VBS space are not modeled, but experimentally derived based on measured gas-particle partitioning, as described in Section 2.3. The 2D-VBS framework is used as a mean to visualize the compound-resolved volatility and oxidation characteristics, following the conceptual structure of Donahue et al. (2011) and Murphy et al. (2012). Its usage is already described in Section 3.3 of the manuscript, where we previously refer to "measured volatilities" and explain the placement of compounds in terms of SVOC and IVOC ranges. This type of representation has been also demonstrated in previous works (Gkatzelis et al., 2018; Kostenidou et al., 2024; Lannuque et al., 2023). This explanation is made
clearer in the revised manuscript (Lines 521-523):

"A two-dimensional space, the 2D-VBS framework (Donahue et al., 2011; Murphy et al., 2012), is
used as a means to visualize the compounds distribution as function of the experimentally
derived volatility (log10Ci*) and the O/C ratio (Fig. 5) or the oxidation state of carbon (OSc) (Fig.
S4)."

4. Conclusions:

A lot of measured result description is shown in the current manuscript. Would be possible for
authors to add more discussion?

The authors already discussed the different results in the relative sections (3.1, 3.2, 3.3). We do
not think the conclusion of a scientific article is the right place to have discussion; for us it is
rather the place to make a summary of the most meaningful results and eventually their
consequences.

5. Reference:

Fan et al., Temperature effects on SOA formation of *n*-dodecane reaction initiated by Cl atoms;
https://doi.org/10.1016/j.atmosenv.2025.121070.

**#reviewer number 2 (RC2)**

General remarks:

In this paper, the effect of temperature and VOC/NOx ratio on SOA formation and partitioning of
individual SOA products from m-xylene and naphthalene OH-oxidation. Experiments are carried
out in an oxidation flow reactor (OFR) and products are identified and quantified using a proton
transfer reaction time-of-flight mass spectrometer (PTR-ToF-MS) coupled to a CHemical Analysis
of aeRosol ONline (CHARON) inlet. Results show that lower temperatures significantly enhance
SOA formation, while lower VOC/NOx ratios reduce it. Gas-phase m-xylene major products are
C3, C5 and C8 whereas particle-product distributions exhibit a progressive increase from C2 to
C8. In contrast, naphthalene products partition more readily into the condensed phase, with C8-
C10 products dominating. Most of the oxidation products from both precursors exhibit a volatility
distribution in the SVOC regime, with fewer in the IVOC regime. A comparison between observed
and estimated volatilities using SIMPOL.1 model reveals systematic deviations for both light
molecules and heavy compounds, suggesting a need for improved predictive models. This is a
nice piece of work. However, the article, the figure and the table have to be ameliorated. In
particularly, there is no uncertainties of measurements, it will be good to add it when it's possible.
Furthermore, some explanation are missing in the manuscript, as the temperature influence, the
model VBS, the comparison with literature (yield and also products). The data presented are of
significant importance and should be published after some minor revisions. For more
explanation, see specific comments.

Specific Comments

**Line 18-19: Define $C_i$\* and SIMPOL in the abstract**

Both terms are shortly described now in the *Abstract*:

*"The decrease in temperature shifts **the effective saturation concentration ($C_i^*$)** towards lower*
*values, though no clear relationship between $C_i^*$ and oxidation state is observed. A comparison*
*between observed and estimated volatilities using **a model based on group contribution***
***method (SIMPOL.1)** reveals systematic deviations for both light molecules and heavy*
*compounds, suggesting a need for improved predictive models."*

And more details are given in the updated manuscript:

*"where $C_i^*$ is known as the effective saturation concentration of species i which describes the gas-*
*particle partitioning behavior of organic compounds and represents the gas-phase concentration*
*of a compound at which it will partition equally between the gas and particle phases under given*
*atmospheric conditions, and is calculated as the reciprocal of $K_{p,i}$ (Donahue et al., 2006, 2011)".*
Extended definition is provided in *Section 2.3, Lines 224-228.*

*"SIMPOL.1, a group contribution method (Pankow and Asher, 2008) which implements a*
*structure-activity relationship method to calculate the subcooled pure liquid vapor pressure by*
*summing the contributions of the subcooled liquid vapor pressures of individual chemical*
*functional groups",* as described in *Section 2.3, Lines 231-234.*

**Line 28**: give the range of vapor pressures for SVOC and IVOC

The vapor pressure along with $C_i^*$ ranges for SVOCs and IVOCs have been included in the main
text based on (Donahue et al., 2012) *(Section 3.3, Lines 520-523)*:

*"The volatilities of the identified compounds are in the range of SVOCs with $C_i^*$ values ranging from*
*0.3 to 300 µg/m³ (vapor pressures approximately $10^{-8}$ to $10^{-5}$ torr) to IVOCs with $C_i^*$ values ranging*
*from 300 to 3 × $10^6$ µg/m³ (vapor pressures approximately $10^{-5}$ to $10^{-1}$ torr) (Donahue et al.,*
*2012)."*

**Line 57-58**: Define OFR system and C*i**

This definition of OFR is added in the *Introduction, Lines 93-94*:

*"The oxidation flow reactor (OFR, Fig. 1) is a continuous-flow reactor that uses substantially*
*elevated oxidant levels to rapidly simulate atmospheric oxidation chemistry (Peng and Jimenez,*
*2020; Kang et al., 2007)."*

$C_i^*$ is defined above in response to a previous comment.

**Line 80**: In the literature, it will be good to check if the temperature and NOx effect has been
already studied for naphthalene and xylene. What are the temperature studied in this article?

Yes, temperature and NOx effects on SOA yields were already reported, but the effects of such
conditions on chemical composition and more specifically the partitioning are not reported for
both species so far. Results of previous studies on SOA yield are presented in Fig. 2 and in section
3.1, where we discussed our results with respect to the literature (Chan et al., 2009; Chen et al.,
2016, 2018; Ng et al., 2007; Song et al., 2007; Takekawa et al., 2003).

The temperatures studied are presented in the paper and are 280 K and 295 K.

**Line 84 :** "their SOA formation potential has been previously demonstrated (Chan et al., 2009;
Chen et al., 2016, 2018; Lu et al., 2024; Ng et al., 2007; Song et al., 2007; Srivastava et al., 2022)".
Some reference are missing for aromatics compounds. For example, it will be good to add the
reference below

Sato, K., Ikemori, F., Ramasamy, S., Iijima, A., Kumagai, K., Fushimi, A., Fujitani, Y., Chatani, S.,
Tanabe, K., Takami, A., Tago, H., Saito, Y., Saito, S., Hoshi, J., & Morino, Y. (2022a). Formation of
secondary organic aerosol tracers from anthropogenic and biogenic volatile organic compounds
under varied NO and oxidant conditions. Atmospheric Environment: X, 14, 100169.
https://doi.org/10.1016/j.aeaoa.2022.100169

The suggested reference has been added, along with several additional relevant references. It
was initially excluded since this study reports chamber experiments at very high concentrations
(in the order of ppm), while we conducted OFR at precursor concentrations < 100 ppb.

**Line 88**: Have you verified that the two compounds are not photolysed by the lamps.

We have verified that neither m-xylene nor naphthalene undergo significant photolysis under the
experimental conditions by comparing the concentration of both precursors when successively
switching UV lights on and off (in absence of $H_2O_2$). This can be easily explained since the UVB
lamps employed (Model UV-B BL SG 1200 DE 40, code 27V00027) have an emission spectrum in
the range 280-350 nm peaking at 310 nm (see figure below), which lies outside the main absorption bands of both compounds (m-xylene and naphthalene absorb predominantly at
wavelengths < 280 nm). Therefore, direct photolysis of the precursors is negligible in our system.
Furthermore the OFR tube filters some of the irradiated light as can be observed below.

[Figure]

Irradiance spectra of UVB lamps with and without quartz tube. The presence of quartz lower the
irradiance peak at 310 nm.

**Line 99**: Have you verify that this H2O2 (which stabilizer is used) don't formed SOA=> Have you
performed blank experiment with H2O2, lamps without the VOC.

Yes, we have performed blank experiments on a daily basis. In fact, impurities may arise from
different parts of the experimental set up, such as milliQ water, OFR walls, cylinder used for NOx,
the ammonium sulphate solution, and as mentioned by the reviewer the stabilizer in $H_2O_2$. To
characterize the effect of such impurities, a blank experiment was conducted prior to each SOA
experiment, using the same conditions in the absence of the studied precursor (either m-xylene
or naphthalene). We did observe a low amount of particles formed under these blank experiments
(typically a 1-2 µg m$^{-3}$), but they could not be attributed exclusively to the $H_2O_2$ stabilizer, as many
potential sources of impurities existed. The products formed during these daily blanks were
quantified and subtracted to the signal of the following experiment. This approach ensures that
the reported SOA yields reflect only the contributions from the aromatic precursor oxidation.

The following is added to the updated manuscript *(Section 2.2, Lines 170-173)*:

*"A blank was conducted prior to each SOA experiment, using the same conditions ($H_2O_2$ flow, NOx*
*concentration, temperature, humidity, etc.), in the absence of the VOC precursor (either m-xylene*
*or naphthalene). The products formed during these daily blanks were quantified and subtracted*
*to the signal of the following experiment."*

**Line 102:** WHAT is the temperature of the kinetic constant used, this is comparable you're your
temperature used during your experiment?

We used the rate constant value at 298 K: $k_{OH+xylene}$ = 2.3 × 10$^{-11}$ cm$^3$ molecules$^{-1}$ s$^{-1}$ and $k_{OH+naphthalene}$
= 2.3 × 10$^{-11}$ cm$^3$ molecules$^{-1}$ s$^{-1}$, for all the experiments of both m-xylene and naphthalene as
reported in Calvert et al. (2015). Following the reviewer's comment, we calculated the rate
constants at 280 K based on Arrhenius equation (see below), yielding $k_{OH+xylene}$ = 2.36 × 10$^{-11}$ cm$^3$
molecules$^{-1}$ s$^{-1}$ and $k_{OH+naphthalene}$ = 2.47 × 10$^{-11}$ cm$^3$ molecules$^{-1}$ s$^{-1}$. The new estimated OH radical
concentrations in the low T experiments are now available in the revised manuscript (Table 1).
The updated rate constants induced only a slight change in the OH concentration not affecting
the discussion.

This paragraph reads as follows in the updated manuscript *(Section 2.1, Lines 133-137)*:

*"For each experiment, the OH radical concentration generated was estimated by fitting the VOC*
*precursor (m-xylene or naphthalene) decay assuming a pseudo first order reaction with OH*
*radicals using temperature-dependent values of the kinetic rate constant as recommended from*
*NIST Kinetics Database. This estimation is also based on the hypothesis that the other reactions*
*with OH do not limit the reaction of the precursor in the first seconds following lamps switching*
*on."*

The calculation of the temperature-dependent OH rate constant is added in the updated SI *(Lines*
*25-36)*:

***"Calculation of the OH rate constant***

*The values of the temperature-dependent OH rate constant are calculated based on Arrhenius*
*equation as follows:*

$$k_{OH}\,(T) = \; A \; \frac{T}{298\,(K)} \; e^{\frac{-Ea}{RT}}$$

*where*
• *$k_{OH}(T)$ is the rate constant at temperature (T) [cm$^3$ molecule$^{-1}$ s$^{-1}$]*
• *T is the temperature [K]*
• *A is the pre-exponential factor*
• *R is the universal gas constant [8.413 J mol$^{-1}$ K$^{-1}$]*
• *Ea is the activation energy [J mol$^{-1}$ ]*

*The specific parameters were (taken from NIST Kinetic Database):*
• *m-xylene: A = 1.66 × 10$^{-11}$, Ea = -964 J mol$^{-1}$*
• *naphthalene: A = 1.05 × 10$^{-12}$, Ea = -7500 J mol$^{-1}$"*

**Line 106** : blank experiment performed. Where comes from the contamination below m/z 61?
Why this m/z?

Background contamination was observed for small ions, including acetic acid at *m/z* 61.03, which
explain why we started the analysis after this specific *m/z*. This background signal was also
observed in blank experiments conducted under identical conditions without precursor injection,
but as stated earlier, it is difficult to identify its origin(s).

Although background subtraction was applied during data processing, species in the low *m/z*
range exhibit some variability (potentially due to ambient laboratory VOCs, instrument
outgassing, milliQ water or the OFR walls), or residuals from previous experiments. These species
are commonly detected by the very sensitive PTR-ToF-MS and can persist even under clean
conditions.

In addition the contribution of these low compounds are not expected to account as major
oxidation products and are reported to be negligible for SOA based on previous CHARON-PTR-
ToF-MS measurements (Leglise et al., 2019; Müller et al., 2017; Piel et al., 2019). As a result, the

*decision to exclude signals below m/z 61 aimed at avoiding overinterpretation of ambiguous low-*
*mass signals.*

**Line 130** : naphthalene is not in the standard solution. What is the uncertainties of the calibration
using the transmittance curve?

*Naphthalene is not reported to undergo fragmentation in the PTR, so the main source of*
*uncertainty stems from the transmission curve fitting. Since naphthalene's mass falls within the*
*calibrated mass range and within the mass range of the VOC included in the standard, its*
*quantification is quite robust. Based on the fit and experimental conditions, we estimate the*
*uncertainty in the calibration to be ± 30 %.*

Line 143: Do you form aerosol during the blank experiment?

*We did answer above.*

Line 155-160: "The AMS data were corrected by collection efficiency (CE) calculated by
comparison to the SMPS (Scanning Mobility Particle Sizer, TSI Classifier model 3082, DMA, TSI
CPC 3776) volume using densities of 1.7 g cm-3 for ammonium sulphate and 1.4 g cm-3 for
organics. The CE values varied from 0.3 for pure ammonium sulphate particles to 0.7 after SOA
formation".

Add reference for THE DENSITY OF AMMONIUM SULFATE AND FOR ORGANICS

THE CE IS THE SAME FOR XYLENE AND NAPHTALENE. And this value does not change with HR.
The line of sampling is dried or not? What is the VALUE OF CE FOR AMMONIUM SULFATE IN THE
LITTERATURE?

*The density used for ammonium sulfate particles (1.7 g cm$^{-3}$) is consistent with values reported*
*in the literature. For example, Freedman et al. (2009) measured a density of approximately 1.77 g*
*cm$^{-3}$ for pure ammonium sulfate particles. For organic aerosols, we applied a density of 1.4 g cm$^-$*
*$^3$, consistent with typical SOA derived from hydrocarbon oxidation. Both Hunter et al. (2014) and*
*Freedman et al. (2009) observed SOA densities in the range of 1.2-1.4 g cm$^{-3}$, depending on SOA*
*composition and mixing state, supporting our chosen value.*

*The collection efficiency (CE) for pure ammonium sulfate particles can be verified at the*
*beginning of each experiment before SOA was formed. The CE was determined individually for*
*each experiment by comparing AMS mass concentrations to SMPS-derived particle volumes*
*using the appropriate density. The CE varied from approximately 0.35 for pure ammonium sulfate*
*to up to 0.7 after SOA formation, consistent with established composition-dependent behavior.*
*The literature value of CE for pure ammonium sulfate particles in AMS measurements is typically*
*~0.25-0.4, as noted in Matthew et al. (2008) and Docherty et al. (2013), in agreement with our*
*experimental observations.*

*Regarding humidity control, the ammonium sulfate seed aerosol was passed through a silica gel*
*dryer prior to injection into the chamber. This step reduced initial water content associated with*
*seed generation but did not actively dry the aerosol stream during sampling. The line with $H_2O_2$*
*was not dried.*

**Figure 1** : Define MFC

*"MFC: mass flow controller."* It has been added in the caption of Fig. 1.

**Line 175** : TSP is the total suspended particulate matter of the aerosol (in µg m-3) as measured
by SMPS. SMPS measured a number and not, you have to apply a density.

It is indeed correct that the SMPS measures number- and volume-based size distributions. In our
analysis, we used the number concentration (particles cm$^{-3}$) from the SMPS and converted it to
mass concentration (µg m$^{-3}$) by assuming spherical particles and applying the standard geometric
volume formula for each size bin. Each calculated particle volume was then multiplied by the
assumed composition-specific density to determine mass. Specifically, the densities used were:
1.0 g cm$^{-3}$ for water, 1.4 g cm$^{-3}$ for organics, and 1.7 g cm$^{-3}$ for ammonium, sulfate and nitrate. The
percentage of each chemical fraction for each experiment was inferred from AMS measurements,
and the corresponding weighted average density was applied to the SMPS volume distribution to
obtain TSP. This approach provided a more accurate representation of the aerosol mass.

This description of the TSP calculation is added in the SI *(Lines 37-45)*.

**Line 191**: Define the term in equation 5 (Clapeyron one)

The definitions of the terms of Eq. 5 are added in the manuscript *(Section 2.3, Lines 243-245)*:

*"where $C^{°}_{i,293}$ is the saturation concentration calculated by Volcalc at T = 293 K, $\Delta H_i$ is the enthalpy*
*of vaporization of species i (computationally predicted values from ChemSpider), and R is the*
*ideal gas constant = 8.314 J mol$^{-1}$ K$^{-1}$."*

Table 1: add the uncertainties in the table for each parameters

Table 1 has been updated and the uncertainties added for each parameter.

**Table 1. List of conducted laboratory experiments and associated conditions, such as OFR temperature,**
**RH, VOC/NO$_x$ ratio, seeds mass and SOA yield.**

| | T | RH | VOC | NO$_x$ | VOC/NO$_x$ | Seeds | $[OH] \times 10^7$ | $\Delta VOC$ | | $\Delta M_0$ | Y |
|---|---|---|---|---|---|---|---|---|---|---|---|
| | K | % | ppbV | ppb | ppbC ppb$^{-1}$ | µg m$^{-3}$ | molecules cm$^{-3}$ | µg m$^{-3}$ | % | µg m$^{-3}$ | % |
| **m-xylene** | 280 ± 1.5 | 75 ± 5 | 74 ± 0.65 | 235 | 2.5 ± 0.1 | 51 ± 1.6 | 3.4 ± 0.5 | 114 ± 3.3 | 34 | 26.4 ± 1.2 | 23.1 ± 1.2 |
| | 280 ± 1.5 | 50 ± 5 | 69 ± 0.70 | 40 | 13.9 ± 0.5 | 35 ± 0.5 | 2.2 ± 0.4 | 95 ± 3.3 | 30 | 26.1 ± 1.9 | 27.5 ± 2.1 |
| | 295 ± 2 | 60 ± 7 | 73 ± 0.66 | 221 | 2.6 ± 0.1 | 30 ± 0.8 | 5.4 ± 0.8 | 155 ± 3.2 | 49 | 12.6 ± 1.4 | 8.1 ± 1.0 |
| | 295 ± 2 | 55 ± 7 | 83 ± 0.74 | 40 | 16.6 ± 0.6 | 34 ± 0.7 | 4.7 ± 0.7 | 153 ± 3.2 | 42 | 21.1 ± 1.4 | 13.8 ± 1.0 |

| naphthalene | | | | | | | | | | |
|---|---|---|---|---|---|---|---|---|---|---|
| 280 ± 1.5 | 40 ± 3 | 57 ± 0.35 | 340 | 1.7 ± 0.1 | 46 ± 0.6 | 2.9 ± 0.2 | 92 ± 2.3 | 29 | 23.3 ± 1.8 | 25.3 ± 2.0 |
| 280 ± 1.5 | 35 ± 3 | 53 ± 0.41 | 62 | 9.3 ± 0.4 | 36 ± 1.0 | 3.2 ± 0.3 | 79 ± 2.3 | 27 | 33.6 ± 6.9 | 42.7 ± 8.8 |
| 295 ± 2 | 40 ± 5 | 53 ± 0.47 | 340 | 1.6 ± 0.1 | 55 ± 1.5 | 3.5 ± 0.3 | 84 ± 2.3 | 28 | 12.6 ± 1.7 | 14.9 ± 2.1 |
| 295 ± 2 | 50 ± 5 | 49 ± 0.41 | 57 | 8.6 ± 0.4 | 65 ± 0.3 | 3.1 ± 0.4 | 75 ± 2.2 | 29 | 13.8 ± 1.6 | 18.3 ± 2.2 |

**Line 244-249**: add reference for all the software used (SPSS software, hysplit...) For hysplit, add
a reference and also the model used (GDAS, backward trajectory...) For example, the reference
can be Stein, A. F., Draxler, R. R., Rolph, G. D., Stunder, B. J. B., Cohen, M. D., and Ngan, F.:
NOAA's HYSPLIT Atmospheric Transport and Dispersion Modeling System, Bulletin of the
American Meteorological Society, 96, 2059–2077, https://doi.org/10.1175/BAMS-D-14-00110.1,
2015.

This comment is not relevant for the present article since it is a laboratory study that does not use
HYSPLIT. We suggest the reviewer check their comments.

**Line 281, Figure 2**. Add the period of measurement before, during and after the Diwali period? In
the figure, it will be good to add also the uncertainties.

This comment is not relevant for the present article.

**Figure 4**: This figure is not necessary. It's impossible to keep this figure like this in the article =>
there is no linear correlation between the concentration and the T, and Humidity. You have really
to redo this figure or mix this figure with 3. You have to remove the linear correlation. For example
in the figure 3, add the temperature and humidity in function of time as the concentration and
highlight different period during the campaign. If you remove this figure, you have also to modify
the text in the manuscript.

This comment is not relevant for the present article.

**Line 412-413**: The HYSPLIT was used to generate 72-hour backward trajectories at various
altitudes above ground level (AGL) for the sampling period, highlighting the origin of air pollutants
at the study site. This text can be also added in the figure caption and also the reference of
HYSPLIT

Line 419-420; Figure 19 explain the backward trajectory, in the figure caption. Explain in the
legend the different backward trajectory, what is the difference between the red and blue one.

This comment is not relevant for the present article.

**LINE 422**: GIVE the equation of AQI?

This comment is not relevant for the present article.

**LINE 462**: add the uncertainties of measurement of elemental characterization

This comment is not relevant for the present article.

**Table 3 and table 4**: it's difficult to follow the results of these two tables. Thanks to add also the
uncertainties. For me, it will be easier to show the results with a chart graph to see more the
similarities and the uncertainties.

This comment is not relevant for the present article (we have Tables 2 and 3).

**Figure 2:** in the case of xylene oxidation at high and low temperature, an inconsistent trend
between the previous studies and this study. You have to explain the inconsistence between this
work and the literature. For the literature, add also the VOC/NOx ratio, the temperature (all the
parameters that can explain the difference). For naphthalene is more consistent with the
literature. It's also difficult to compare the yield with two points.

We would like to respectfully clarify that our observed temperature trend for m-xylene SOA yields
is consistent with prior studies when considering the full experimental context, specifically
VOC/NO$_x$ ratios, humidity, seed presence, and initial precursor concentrations. These
comparisons are discussed in *Section 3.1, Lines 252-304* of the updated manuscript, with
supporting data in Figure 2a.

At 295 K under high-NOx conditions, we report a SOA yield of ~8 % (filled red square), which is
consistent with prior measurements by Ng et al. (2007) and Chen et al. (2018). Ng et al. used AS
seed particles, HONO as the OH precursor, and xylene concentrations of 42–171 ppb. Chen et al.
used a similar VOC/NOx ratio with slightly lower precursor levels (44–59 ppb) and no seed
particles.

The observed increase in SOA yield at lower temperature (from 8 % at 295 K to 23 % at 280 K)
aligns with the trend reported by Takekawa et al. (2003), who found yields rising from 6 % to 13 %
over a 20 K drop.

For low NO$_x$ conditions, our 295 K yield of 14 % (empty red square) falls within the range reported
by Song et al. (2007) (12-30 %), who used lower xylene concentrations (39-52 ppb).

In the manuscript, we also discuss additional parameters, such as humidity effects (Li et al.,
2022) and seed presence (Lambe et al., 2015), that can explain variability across studies.

Finally, while our temperature comparison is based on two data points, they were deliberately
chosen to span an atmospherically relevant range (280-295 K) and are sufficient to reveal a clear
and interpretable temperature dependence under controlled conditions.

[Figure]

**Figure 2. SOA yields at 295 K and 280 K as function of organic aerosol mass formed for (a) m-xylene and (b) naphthalene in comparison with previous studies. Filled markers correspond to high NO$_x$ conditions, open markers to low NO$_x$.**

**Line 255**: why you can't measure glyoxal and why is an important product?

Glyoxal is as an important product of aromatic oxidation (Nishino et al., 2009; Volkamer et al., 2001), but as noted in the manuscript *(Section 3.2.1, Lines 311-313)*, glyoxal's signal was masked due to the significantly higher intensity of the acetone peak, making it not possible to resolve glyoxal accurately. Glyoxal ($C_2H_2O_2$) has an *m/z* value at 59.01, which is very close to that of acetone ($C_3H_6O$) at *m/z* 59.05. Additionally, glyoxal has a very low proton affinity, resulting in poor ionization efficiency in the PTR-ToF-MS and further reducing its detectability.

**Line 338**: It will be good to add some discussion on chemical composition under the different environmental conditions and also to compare your work with the work of Sato, 2022 for tracer from oxidation of naphthalene and xylene. For naphthalene, do you see the succinic acid?

We did not necessarily focus the discussion on one study since several others exist on SOA formation from both naphthalene and m-xylene. Among the different compounds studied by Sato et al. (2022), only nitrophenol and nitronaphthol were detected in our study. This is mainly explained by the experimental conditions that highly differed between the two studies . The following table summarize the experimental conditions:

| Parameter | Our Study | Sato et al. (2022) |
|---|---|---|
| Temperature | 280 K and 295 K | ~298 K (Room Temperature) |
| RH | 35-74 % | <1 % (very dry conditions) |
| Oxidant Source | H$_2$O$_2$ photolysis | for high NO$_x$: methyl nitrite CH$_3$ONO photolysis; for low NO$_x$: H$_2$O$_2$ |
| Photolysis Wavelength | UVB lamps (280-350 nm, $\lambda_{max}$ = 310 nm) | Xenon arc lamps (broadband 250-1000 nm, strong in 300-400 nm) |
| NO$_x$ Introduction | NO$_2$ added via gas cylinder | for high NO$_x$: CH$_3$ONO (generates NO), for low NO$_x$: NO$_2$ |

| Initial NO_x Levels | 40-340 ppb | High NO_x (~50-470 ppb), Low NO_x (<1 ppb) |
|---|---|---|
| VOC/NO_x Ratio | 1.6-16.6 ppbC/ppb | 14-139 ppbC/ppb |
| [OH] | $2.3\text{-}5.4 \times 10^7$ molecules cm$^{-3}$ | $0.74\text{-}24.9 \times 10^6$ molecules cm$^{-3}$ |
| Residence Time | ~8 minutes | Likely > 30 minutes (typical chamber conditions) |
| Seed Aerosol | ammonium sulfate | none |
| Detection Method | Online CHARON-PTR-ToF-MS | Offline UPLC-ESI-MS/MS and GC-MS |
| Nitroaromatic Products | Not detected or at limit of detection | Detected and identified |

The low level of nitrated aromatic compounds in our system is in agreement with previous studies with closer conditions compared to that of Sato et al. (2022), where $CH_3ONO$ is used as OH radical precursor, which introduces much larger NO concentrations.

Finally, the use of online CHARON-PTR-ToF-MS, while advantageous for real-time detection of oxygenated VOCs, SVOCs and IVOCs, is less sensitive to very low-volatility, low-proton-affinity compounds such as nitrated compounds, which were analyzed offline by Sato et al. using offline LC-MS and GC-MS techniques. These factors together likely account for the non-detection or near-LOD levels of nitroaromatic products in our study.

The discussion on naphthalene tracers has been added in the updated manuscript *(Section 3.2.2, Lines 472-478)*, and the work of Sato et al. (2022) is included among the literature cited.

As expected, higher contribution of organonitrate products has been observed for high NO_x conditions, but the fraction of organonitrates is moderate in all conditions. To support this discussion, we updated Figs. 3, 4, S1, and S4 (Fig. S4 previously denoted as Fig. S3) to identify the contribution of N-compounds, and added new Figs. S2 and S5 in the Supplement (also presented below). We have also expanded our discussion to include NO_x effects on chemical composition in the revised manuscript: *(Section 3.2.1, Lines 334-341, 380-386; Section 3.2.2, Lines 443-446, 468-478).*

[revised manuscript text omitted]

**Line 464:** check if you can find other article for anthropogenic compounds and the effect of volatilities. For example, you can check some article for alkanes, as dodecane (Lamkaddam et al).

Lamkaddam Houssni, Aline Gratien, Edouard Pangui, Marc David, F. Peinado, Jean-Michel Polienor, Murielle Jerome, Mathieu Cazaunau, Cecile Gaimoz, Benedicte Picquet-Varrault, I. Kourtchev, M. Kalberer, Jean-Francois Doussin, Role of Relative Humidity in the Secondary Organic Aerosol Formation from High-NOx Photooxidation of Long-Chain Alkanes: n-Dodecane Case Study, ACS Earth Space Chem. 2020, 4, 2414–2425, 4, 2414–2425, https://doi.org/10.1021/acsearthspacechem.0c00265, 2020

Lamkaddam Houssni, Aline Gratien, Edouard Pangui, Mathieu Cazaunau, Benedicte Picquet-Varrault, Jean-Francois Doussin, High-NOx Photooxidation of n-Dodecane: Temperature Dependence of SOA Formation, *Environmental Science and Technology*, 51, 192-201, 10.1021/acs.est.6b03821, 2017

In our study, we observed a clear temperature dependence in gas-particle partitioning, with lower temperatures favoring increased partitioning to the particle phase, consistent with classical volatility-based behavior. This finding supports the idea that volatility remains a key driver of SOA formation for certain aromatic precursors, such as m-xylene and naphthalene. In contrast, Lamkaddam et al. (2016, 2020) investigated SOA formation from n-dodecane under high $NO_x$ conditions and reported only a weak temperature sensitivity. Their results suggest that SOA from long-chain alkanes may be dominated by extremely low-volatility products, and also suggest that changes in the types of products formed at different temperatures may compensate for volatility effects, resulting in relatively stable SOA yields. This comparison underscores the chemical specificity of SOA volatility behavior among anthropogenic precursors, with alkanes and aromatics potentially exhibiting distinct oxidation pathways and gas-particle partitioning dynamics.

While Lamkaddam et al. (2020) report some trends in their 2D-VBS plot, such as decreasing $\log_{10}C_i^*$ with increasing OSc for CHON and CHNOS species in the low volatility regime ($\log_{10}C_i^* < 0$), this relationship does not persist at higher volatilities ($\log_{10}C_i^* > 0$), where OSc spans a wide range without a consistent trend. This further supports our observation that for anthropogenic VOCs (Fig. S6 in the updated SI), that volatility and oxidation state are not uniformly correlated, particularly for more volatile oxidation products in the SVOC and IVOC ranges.

We added in the updated manuscript *(Introduction, Lines 79-80)*:

*"Lamkaddam et al. (2016) found a weak temperature sensitivity in SOA formation from n-dodecane, suggesting that extremely low-volatility products and compensating shifts in product types may offset volatility effects."*

**Figure 5**: explain the color SVOC and IVOC and VBS framework in the figure caption and also in the manuscript

The light green shading correspond to SVOCs ($0 < \log_{10}C_i^* < 2.5$) while the light blue shading corresponds to IVOCs ($2.5 < \log_{10}C_i^* < 5$). The explanation of those ranges and the 2D-VBS framework is made clearer in the revised manuscript in the caption of Fig. 5, and in the manuscript as follows *(Section 3.3, Lines 519-523)*:

*"The volatilities of the identified compounds are in the range of SVOCs with $C_i^*$ values ranging from 0.3 to 300 µg m$^{-3}$ (vapor pressures approximately $10^{-8}$ to $10^{-5}$ torr) to IVOCs with $C_i^*$ values ranging from 300 to $3 \times 10^6$ µg m$^{-3}$ (vapor pressures approximately $10^{-5}$ to $10^{-1}$ torr) (Donahue et al., 2012), indicated by the light green ($\log_{10}C_i^*$ from 0 to 2.5) and light blue ($\log_{10}C_i^*$ from 2.5 to 5) background shading, respectively."*

 **#reviewer number 3 (RC3)**

General comments

Shahin et al. used an oxidation flow reactor (OFR) to generate m-xylene and naphthalene aerosol
particles at 280 K and 295 K under low and high $NO_x$ conditions. They used a suite of real-time
instruments to investigate the gas- and particle-phase products. The method is overall
technically sound, but the current analysis still lacks details. The main drawback of the work is
that it reads like a measurement report, especially for sections 3.1 and 3.2. The associated
content should be shortened and condensed for readability. It is hard to grab the take-home
message from the study with good experimental design. Efforts must be made to address the
following comments and highlight the novelty before the work can be considered for publication.

Major Comment

1. Introduction: Previous researchers have conducted extensive studies on the effects of
temperature on SOA yield and product volatility, though not all of them involve aromatics,
their research works are also of great reference (Svendby et al., 2008; Clark et al., 2016; Price
et al., 2016; Li et al., 2019; Li et al., 2020; Deng et al., 2021; Lannuque et al., 2023; Fan et al.,
2025). The authors should give a good summary of existing studies in the introduction.
Additionally, the authors should how this work differs from Lannuque et al. (2023).

The above mentioned studies are of course interesting but are dealing with precursors other than
aromatics (terpenes, dodecane, amines, etc.), and include different oxidants (Cl atoms, ozone,
nitrate radicals). We include these studies as a general understanding of the effect of
temperature on SOA yields, and then focus more on the temperature effect on aromatics to
match the present study goal. We now propose to include the following paragraph to the revised
version of the manuscript *(Introduction, Lines 69-84)*:

*"It is well recognized that SOA yields increase at lower temperatures, a trend consistently*
*reported for both terpenes and isoprene (Svendby et al., 2008; Virtanen et al., 2019; Deng et al.,*
*2021; Clark et al., 2016), aromatics (Svendby et al., 2008; Lannuque et al., 2023), alkanes (n-*
*dodecane in Li et al., 2020 and Fan et al., 2025) and amines (Price et al., 2016). This behavior is*
*attributed primarily to the decrease of the vapor pressures of the compounds, displacing the*
*equilibrium towards the particle phase. However, the impact of temperature on SOA composition*
*is not fully understood, and seems to depend on other experimental conditions (precursor, seed*
*acidity, etc.). For biogenic precursors, previous studies reported more oligomer formation at*
*lower temperatures (Li et al., 2020; Fan et al., 2025), driven by increased SVOC partitioning and*
*condensed-phase reactions, while others observed the opposite, attributing higher-temperature*
*oligomerization to radical or acid-catalyzed reactions (Clark et al., 2016; Deng et al., 2021; Price*
*et al., 2016). Additionally, Li et al. (2019) highlighted how lower temperatures increase SOA*
*viscosity, suppressing evaporation and favoring retention of low-volatility species. Conversely,*
*Lamkaddam et al. (2016) found a weak temperature sensitivity in SOA formation from n-*
*dodecane, suggesting that extremely low-volatility products and compensating shifts in product*
*types may offset volatility effects. Regarding aromatic compounds, only Lannuque et al. (2023)*
*investigated the effect of temperature on SOA chemical composition at molecular level, showing*
*a general agreement of product distribution between 280 K and 295 K. Thus, more studies on*
*different aromatic precursors and experimental conditions are needed to complete our*
*understanding of the temperature effect on SOA formation from aromatic precursors."*

To our knowledge, this study is the first to investigate the gas-particle partitioning behavior of SOA
formed from m-xylene and naphthalene oxidation under varying temperature conditions.

While our study and Lannuque et al. (2023) share a similar experimental framework (OFR +
CHARON-PTR-ToF-MS), they differ on the OH radical precursor and type of VOC. We used $H_2O_2$
while Lannuque et al. (2023) used isopropyl nitrite. In addition, we investigate two different
aromatic compounds and different VOC/$NO_x$ regimes, while Lannuque et al. (2023) only focused
on toluene in low VOC/$NO_x$ (e.g. high NOx conditions). Lannuque et al. included a comprehensive
modeling using a dynamic model (SSH-aerosol) while our study is only experimental.

2. OFR: the overall input flow is 2.4 lpm, and thus, the residence time is fairly long. I am
concerned about the significant vapour and particle losses inside the OFR. These artifacts
will potentially affect the observed SOA yield and products. However, there is no discussion
about how the results are biased by the vapour and particle losses. I also wonder if any loss
correction has been applied to the results.

The authors conducted different experiments to characterize both gas and particle losses in the
OFR. Particle wall losses at experimental diameters (seeds) were investigated using ammonium
sulfate seeds of 200 nm electrical mobility diameters (generated using an atomizer associated
with a classifier). Particle wall losses have been calculated as the difference of particle
concentration between the inlet and outlet of the OFR and were observed to be between 10 ± 5
%. Regarding gas phase, the losses were evaluated for the two precursors by considering their
concentration at inlet and outlet. For m-xylene losses were around 5 %, and naphthalene losses
typically of 10-15 %. The losses of gas phase precursor and seed particles were controlled daily
before running each SOA experiment. These losses are in the range of reported values by
Lannuque et al. (2023), even if the residence time is lower here (8 min in this study, 13 min in
Lannuque et al. (2023)). We did not carry out the evaluation of wall losses for the reaction
products, but modeling tests carried out by Lannuque et al. (2023) showed that considering both
precursors and reactions products, wall losses introduced a 10-15 % deviation on the SOA yield.
For m-xylene, we can reasonably assume lower losses as the residence time is shorter and the
flow tube has a larger inner diameter. For naphthalene, it is probable that the wall losses were
higher than that of m-xylene. We finally add precisions about the estimation of losses that we
conducted in the methodology section *(Section 2.1, Lines 145-152)*:

*"In this configuration, particle losses (or its transmissions through the OFR) were estimated by*
*comparing the concentration of seed particles at the inlet and outlet of the OFR, when generating*
*seed at 200 nm electrical mobility diameters. These losses were daily checked prior to each*
*experiment and were in the range 10 ± 5 %. In addition, precursor losses were estimated to be*
*around 5 % for m-xylene and 10-15 % for naphthalene. Losses of gaseous products generated*
*during SOA experiments were not experimentally evaluated. Lannuque et al. (2023) showed, in a*
*toluene SOA experiment, that wall losses introduced a 10-15 % deviation on the SOA yield when*
*considering both precursors and reactions products. For m-xylene, we can reasonably assume*
*lower losses as the residence time is shorter and the flow tube has a larger inner diameter. For*
*naphthalene, it is probable that the wall losses were higher than that of m-xylene."*

3. The way to generate OH differs from that used in the widely used PAM. Some levels of simple
model simulation should be provided to understand the chemistry inside the OFR and how
$NO_x$ affects oxidation in the context of used OFR. This can be done by the KimSim model
developed by Peng and Jimenez (2019).

We choose to generate OH radicals at a higher wavelength than traditionally in the PAM chamber
to somehow mimic environmental conditions (See the emission spectra of the corresponding
UVB lamps below). Efficient $H_2O_2$ photolysis was ensured by using 6 lamps around the OFR, and
adjusting the $H_2O_2$ flow. A set of test experiments using different $H_2O_2$ flows has been conducted
prior to SOA formation experiments to evaluate OH concentration generated, calculated using a
fit of the pseudo first order decay of the precursor.

$NO_x$ may play an important role in shaping oxidation chemistry depending on experimental
conditions. We explicitly compared the chemical composition of gas and particle phase products
in environmentally relevant low $NO_x$ and high $NO_x$ conditions, allowing us to evaluate $NO_x$-
dependent product distributions directly (see answer to comment #4 below). No major change
was observed in the products distribution (details are given in the answer below). Also, the
estimated OH did not vary a lot when comparing high $NO_x$ and low $NO_x$ conditions.

[Figure]

4. Low $NO_x$ experiments: One variable in the experiment is the $NO_x$ However, there is no
discussion about the results under low-$NO_x$ conditions from section 3.2 onward. It will be
interesting to see how $NO_x$ affects the chemical distribution and volatility distribution.

As expected, higher contribution of organonitrate products has been observed for high $NO_x$
conditions, but the fraction of organonitrates is moderate in all conditions. To support this
discussion, we updated Figs. 3, 4, S1, and S4 (Fig. S4 previously denoted as Fig. S3) to identify the
contribution of N-compounds, and added new Figs. S2 and S5 in the Supplement (also presented
below). We have also expanded our discussion to include $NO_x$ effects on chemical composition
in the revised manuscript: *(Section 3.2.1, Lines 334-341, 380-386; Section 3.2.2, Lines 443-446,*
*468-478).*

[revised manuscript text omitted]

5. AMS: What was the use of the AMS data in this study? The author should consider including the bulk elemental composition, oxidation states and organonitrate fraction based on the AMS data.

We used the AMS with the unique purpose to evaluate the total SOA mass formed and compare it to the CHARON results, to track CHARON efficiency of OA recovery. As our major interest was to investigate partitioning of individual compounds and describe the molecular level composition of the aerosol that could be provided by the CHARON-PTR-ToF-MS. This is also why we did not quantify organonitrates with the AMS but we identified specific organonitrate compounds using CHARON-PTR-ToF-MS confirming their presence, as presented in Tables 2 and 3. A short analysis of the AMS data shows that photochemistry enhanced nitrate (nitric acid) that condense onto ammonium sulphate particles, and when the organic precursor is added we see a small variability of the $NO^+/NO_2^+$ ratio around 3-4 % for the low $NO_x$ experiments to 8-10 % for the high $NO_x$

experiment. This small variation is somehow in agreement with the PTR-MS CHARON findings.

6. Section 3.4: The authors gave a very good explanation about the differences between the model results and the observational data. But authors should also discuss how temperature and VOC/$NO_x$ ratios affect these differences.

+ Minor Comment #15: Line 514-515: What is the meaning of "temperature-induced volatility changes for highly oxygenated compounds"? In Fig. S5, it seems that at higher temperatures (295K), the discrepancy between the observed and calculated values was smaller than that in 280K. How did the temperature affect the discrepancy?

We have combined the responses to Major Comment 6 and Minor Comment 15, as they address closely related aspects concerning the influence of temperature and VOC/$NO_x$ ratios on the discrepancies between model results and observations.

Figures 6 and S7 show the comparison of estimated versus experimental $\log_{10}C_i^*$ values under high $NO_x$ conditions at 280K and 295 K, respectively, for both m-xylene and naphthalene. The results at 295 K closely match those at 280 K, with no significant difference in the distribution.

Additionally, we have added Fig. S8 to present the same analysis under low $NO_x$ conditions at 295

K, which similarly shows no notable deviation from the trends observed at high $NO_x$ (Fig. S7). Even if differences are visible for some individual compounds, the general trend observed for the ensemble of the data points seems to be very similar in two $NO_x$ regimes. This indicates that neither temperature nor $NO_x$ regime substantially explain the model-observation discrepancy in our dataset.

We have combined below Figs. 6, S7, and S8 for easier visualization and comparison.

[Figure]

To clarify, "temperature-induced volatility changes" refers to the degree to which a compound's
volatility changes as temperature shifts from 280 K to 295 K. This is quantified in Fig.5 which
represents the difference between experimental $\log_{10}C_i^*$ at 280 K and 295 K (i.e. $\Delta\log_{10}C_i^*$) against
the C number.

Regarding the temperature effect on the discrepancy between the experimental and the
estimated $\log_{10}C_i^*$ values, we have added a new figure (Fig. S9), which presents $\Delta\log_{10}C_i^*$, the
difference between estimated and experimental $\log_{10}C_i^*$ values for the oxidation products of m-
xylene and naphthalene as a function $m/z$, at 295 K (magenta) and 280 K (light violet). The
discrepancies at both temperatures are quite close to each other. Overall, SIMPOL'S bias in
predicting volatility for oxidation products is not strongly affected by temperature.

[Figure]

**Figure S9. $\Delta\log_{10}C_i^*$ ($\log_{10}C_i^*$ Estimated minus $\log_{10}C_i^*$ Experimental; y-axis) versus m/z (x-axis) for (a) m-xylene and**
**(b) naphthalene oxidation products under high NOₓ conditions, at 295 K (magenta) and 280 K (light violet).**

This discussion is added in the updated manuscript *(Section 3.4, Lines 608-620)*:

*"Figure S7 shows the comparison of estimated versus experimental $\log_{10}C_i^*$ values at 295 K under*
*high NOₓ conditions for both m-xylene and naphthalene. These results closely match those at 280*
*K (Fig. 6), with no significant difference in the distribution. Figure S8 presents the same analysis*
*under low NOₓ conditions at 295 K, which similarly shows no notable deviation from the trends*
*observed at low NOx (Fig. S7). This indicates that neither temperature nor NOₓ regime*
*substantially explain the model-observation discrepancy in our dataset. Temperature-induced*
*volatility changes are quantified in Figs. 5c,d which represents the difference between*
*experimental $\log_{10}C_i^*$ at 280 K and 295 K (i.e. $\Delta\log_{10}C_i^*$) against the C number. The temperature*
*effect on the discrepancy between the experimental and the estimated $\log_{10}C_i^*$ values is*
*presented in Fig. S9. The values of $\Delta\log_{10}C_i^*$, the difference between estimated and experimental*
*$\log_{10}C_i^*$, for the oxidation products of m-xylene (Fig. S9a) and naphthalene (Fig. S9b) is plotted as*
*a function m/z, at 295 K and 280 K. The discrepancies at both temperatures are quite close to each*
*other. Overall, SIMPOL.1's bias in predicting volatility for oxidation products is not strongly*
*affected by temperature. This consistent discrepancy with temperature and NOₓ confirmed that*
*partitioning of organic compounds is more complex than the simple evaporation of compounds*
*isolated, and that surface interactions and bulk chemical reactions are potential important*
*drivers affecting partitioning equilibria."*

Minor Comment

1. Lines 63-65: "In other studies, SOA yields increase up to 60% under elevated levels of NO$_x$ .......(Srivastava et al., 2023; Zhu et al., 2021)". Although Zhu et al., (2021) mentioned "Nevertheless, the maximum concentration of p-xylene SOA ..., which was 60% higher than that without NO$_2$ ......", it only is related to SOA mass concentrations but not SOA yield. In addition, in Srivastava et al. (2023), though higher SOA yield was found under high NO$_x$ conditions in comparison with low NO$_x$ conditions, the ratios of VOC/NO$_x$ were not known. I suggested if the authors wanted to express the SOA yield increased with VOC/NO$_x$, please include extra reference.

We thank the reviewer for pointing out the error that has been corrected. Accordingly, we have revised the manuscript to reflect the current understanding in the literature *(Introduction, Lines 52-62)*:

*"The NO$_x$ level has been shown to have significant yet possibly contrasting effects on SOA formation (Sarrafzadeh et al., 2016; Liu et al., 2024; Chan et al., 2009; Ng et al., 2007; Qi et al., 2020; Song et al., 2005; Zhu et al., 2021). Some studies have reported a decrease in SOA yield under high NO$_x$ conditions which can be explained by the termination reactions of NO with the peroxy (RO$_2$ and HO$_2$) and OH radicals, limiting the formation of lower volatility compounds (Chan et al., 2009; Ng et al., 2007; Song et al., 2005). Nonetheless, Zhu et al. (2021) reported an increase in SOA mass under elevated levels of NO$_x$ compared to NOx free experiments. Enhanced SOA formation under high NO$_x$ has also been linked to the formation of organic nitrates and the isomerization of alkoxy radicals into low-volatility products, or related to a change in OH concentration due to the presence of NO$_x$ (Ng et al., 2007; Schwantes et al., 2019; Srivastava et al., 2023; Shi et al., 2022). Under low NO$_x$ conditions, RO$_2$ radicals primarily react with HO$_2$ to form low-volatility organic hydroperoxides, which contribute to new particle formation and increase SOA mass (Xu et al., 2014; Zhao et al., 2018). These findings illustrate that NO$_x$ is capable of both inhibiting and promoting SOA formation depending on the NO$_x$ regime."*

2. Lines 103– 106: How was the OH radical concentration determined?

We now mention in the paper *(Section 2.1, Lines 133-137)*:

*"For each experiment, the OH radical concentration generated was estimated for each experiment by fitting the VOC precursor (m-xylene or naphthalene) decay assuming a pseudo first order reaction with OH radicals using temperature-dependent values of the kinetic rate constant as recommended from NIST Kinetics Database. This estimation is also based on the hypothesis that the other reactions with OH do not limit the reaction of the precursor in the first seconds following lamps switching on."*

Following the reviewer's 2 comment, we calculated the rate constants at 280 K based on Arrhenius equation (see below), yielding $k_{OH+xylene}$ = 2.36 × 10$^{-11}$ cm$^3$ molecules$^{-1}$ s$^{-1}$ and $k_{OH+naphthalene}$ = 2.47 × 10$^{-11}$ cm$^3$ molecules$^{-1}$ s$^{-1}$. The new estimated OH radical concentrations in the low T experiments are now available in the revised manuscript (Table 1). The updated rate constants induced only a slight change in the OH concentration not affecting the discussion. We thus did not modify the text following this slight modification of OH radical concentration.

The calculation of the temperature-dependent OH rate constant is added in the updated SI *(Lines 25-36)*:

*"**Calculation of the OH rate constant**

*The values of the temperature-dependent OH rate constant are calculated based on Arrhenius*
*equation as follows:*

$$k_{OH}(T) = A \frac{T}{298\,(K)}\, e^{\frac{-Ea}{RT}}$$

*where*
• *$k_{OH}(T)$ is the rate constant at temperature (T) [$cm^3$ molecule$^{-1}$ s$^{-1}$]*
• *T is the temperature [K]*
• *A is the pre-exponential factor*
• *R is the universal gas constant [8.413 J mol$^{-1}$ K$^{-1}$]*
• *Ea is the activation energy [J mol$^{-1}$ ]*

*The specific parameters were (taken from NIST Kinetic Database):*
• *m-xylene: A = 1.66 × 10$^{-11}$, Ea = -964 J mol$^{-1}$*
• *naphthalene: A = 1.05 × 10$^{-12}$, Ea = -7500 J mol$^{-1}$"*

3. Line 119: Does the activated charcoal affect the collection of organic aerosols?

The potential influence of the activated charcoal denuder on organic aerosol collection has been
thoroughly addressed in Eichler et al. (2015). They show that the denuder removes most particles
with a mobility diameter smaller than 50 nm (< 20 % transmission) as they rapidly diffuse to the
denuder surfaces. Particle transmission steeply increases in the 50-100 nm size range. The
transmission efficiency for 100 nm particles is 75-80 %.Then transmission efficiency exceeds 90
% for particles in the 200-750 nm diameter range, which includes the 200 nm seed particles used
in our study.

4. Line 121: Why 150 deg C was the set temperature of the TD? Is it high enough to vaporise all
compounds in the particles?

Yes, 150 °C is enough to vaporize all the organic compounds from particle phase since the
thermodenuder works under reduced pressure (<10 mbar) as shown previously (Leglise et al.,
2019; Müller et al., 2017). Vacuum conditions lower boiling points, enabling desorption of
S/IVOCs at moderate temperatures.

5. Line 135: How was diiodobenzene introduced into the PTR? Was it done by a permeation
tube?

Diiodobenzene is introduced directly into the drift tube as an internal standard provided by the
manufacturer provided by the manufacturer via a permeation tube connected on the drift, the
reduced pressure and constant temperature in the drift ensuring constant diffusion (PerMaSCal®,
1,3-diiodobenzene, [$C_6H_4I_2$]H$^+$ at *m/z* 330.85 and $C_6H_5I^+$ at *m/z* 203.94, T = 55 °C). This detail has
been added in the manuscript *(Section 2.2, Lines 180-181)*:

*"The peaks at m/z 21.022 ($H_3{}^{18}O^+$), m/z 330.847 corresponding to diiodobenzene ($C_6H_5I_2{}^+$) and its*
*fragment at m/z 203.943 ($C_6H_5I^+$) are used to recalibrate the mass scale (PerMaSCal® internal*
*standard, 1,3-diiodobenzene, T = 55 °C)."*

6. For the yield calculation, did you use the organic mass from SMPS or AMS?

For the SOA yield calculation, we used the organic aerosol mass measured by the AMS as its
chemical resolution is essential for isolating the organic fraction relevant to secondary organic
aerosol formation. SMPS measurements include seeds in addition to SOA.

7. Eq 5: Where did you take the $\Delta H$ values?

The $\Delta H$ values were taken from ChemSpider. These are computationally predicted values, as
experimental thermodynamic data are not reported for many of our oxidation products in
standard databases (such as NIST). This detail has been added alongside the definition of $\Delta H$
*(Section 2.3, Lines 243-244)*:

*"$\Delta H_i$ is the enthalpy of vaporization of species i (computationally predicted values from*
*ChemSpider)".*

8. Table 1: VOC and delta VOC are shown in different units. Could you add another column for
the consumed percentage of VOC?

Table 1 has been revised and the additional column has been added.

**Table 1. List of conducted laboratory experiments and associated conditions, such as OFR temperature, RH, VOC/NO$_x$**
**ratio, seeds mass and SOA yield.**

| | T | RH | VOC | NO$_x$ | VOC/NO$_x$ | Seeds | $[OH] \times 10^7$ | $\Delta VOC$ | | $\Delta M_0$ | $Y$ |
|---|---|---|---|---|---|---|---|---|---|---|---|
| | K | % | ppbV | ppb | ppbC ppb$^{-1}$ | µg m$^{-3}$ | molecules cm$^{-3}$ | µg m$^{-3}$ | % | µg m$^{-3}$ | % |
| **m-xylene** | 280 ± 1.5 | 75 ± 5 | 74 ± 0.65 | 235 | 2.5 ± 0.1 | 51 ± 1.6 | 3.4 ± 0.5 | 114 ± 3.3 | 34 | 26.4 ± 1.2 | 23.1 ± 1.2 |
| | 280 ± 1.5 | 50 ± 5 | 69 ± 0.70 | 40 | 13.9 ± 0.5 | 35 ± 0.5 | 2.2 ± 0.4 | 95 ± 3.3 | 30 | 26.1 ± 1.9 | 27.5 ± 2.1 |
| | 295 ± 2 | 60 ± 7 | 73 ± 0.66 | 221 | 2.6 ± 0.1 | 30 ± 0.8 | 5.4 ± 0.8 | 155 ± 3.2 | 49 | 12.6 ± 1.4 | 8.1 ± 1.0 |
| | 295 ± 2 | 55 ± 7 | 83 ± 0.74 | 40 | 16.6 ± 0.6 | 34 ± 0.7 | 4.7 ± 0.7 | 153 ± 3.2 | 42 | 21.1 ± 1.4 | 13.8 ± 1.0 |
| **naphthalene** | 280 ± 1.5 | 40 ± 3 | 57 ± 0.35 | 340 | 1.7 ± 0.1 | 46 ± 0.6 | 2.9 ± 0.2 | 92 ± 2.3 | 29 | 23.3 ± 1.8 | 25.3 ± 2.0 |
| | 280 ± 1.5 | 35 ± 3 | 53 ± 0.41 | 62 | 9.3 ± 0.4 | 36 ± 1.0 | 3.2 ± 0.3 | 79 ± 2.3 | 27 | 33.6 ± 6.9 | 42.7 ± 8.8 |
| | 295 ± 2 | 40 ± 5 | 53 ± 0.47 | 340 | 1.6 ± 0.1 | 55 ± 1.5 | 3.5 ± 0.3 | 84 ± 2.3 | 28 | 12.6 ± 1.7 | 14.9 ± 2.1 |
| | 295 ± 2 | 50 ± 5 | 49 ± 0.41 | 57 | 8.6 ± 0.4 | 65 ± 0.3 | 3.1 ± 0.4 | 75 ± 2.2 | 29 | 13.8 ± 1.6 | 18.3 ± 2.2 |

9. Line 214: the description of "Lower initial levels of the VOC precursor increase the amount of S/IVOCs (Chen et al., 2019)" is confusing. In the reference, it only showed the decrease in the normalized signal intensity of m/z range from 50 to150 when initial toluene concentration increases from 37 ppb to 690 ppb. In addition, the reference also mentioned: "the concentrations of C7H8O increased as a function of the initial toluene concentration". And in the reference, how many species measured by the PTR could be classified as S/IVOCs is deliberative. The authors need to be more precise when expression.

*The corresponding paragraph has been revised according to the reviewer comment, as following (Section 3.1, Lines 262-270):*

*"For m-xylene (Fig. 2a), the SOA yield at 295 K under high $NO_x$ conditions is approximately 8 % (filled red square), in line with values reported by Ng et al. (2007) and Chen et al. (2018). Chen et al. (2018) reported yields with higher VOC/$NO_x$ ratios, slightly lower precursor concentrations (44-59 ppb), and without using seeds. The absence of seed particles reduces the available surface area for condensation of oxidized products (Lambe et al., 2015). Ng et al. (2007) conducted experiments with AS seeds as in our study, but using nitrous acid (HONO) as OH radical precursor, with varying initial xylene concentrations (from 42 to 171 ppb) which may explain SOA yield variability (from 3 to 8 %). The effect of higher initial levels of the VOC precursor has been shown to reduce aerosol formation (and amount of reacted precursor) and the formation of oxidation products, especially those with m/z > 110 da (Chen et al., 2019). It may be due to the competition between IVOC reactions with OH that produces LVOC/ELVOC, and precursor oxidation. In our case, the initial m-xylene concentration was kept similar between experiments to isolate the effect of temperature and $NO_x$ on SOA yield and chemistry."*

10. Section 3.2: The experiment was conducted under high $NO_x$ Is there any reason there is no N-contained products observed in the m-xylene photooxidation experiments?

We do not understand the point of the reviewer since some N-containing compounds have been detected following m-xylene oxidation (nitro toluene, nitro xylene, dimethyl nitrophenol, nitrocresol and PAN fragment), as presented in Table 1 and discussed in *Section 3.2, Lines 375-383* (updated manuscript). Their concentration is low, as discussed earlier, in agreement with previous studies.

11. Line 254-256: could the author describe how to calculate the carbon balance in SI?

The following description has been added to the SI *(Lines 8-24)*:

*"**Calculation of the carbon balance***

*The carbon balance was calculated by comparing the total carbon content of reaction products to the amount of carbon from the reacted precursor, with all quantities expressed in units of ppbC. First, the amount of precursor carbon reacted was determined by:*

$$reacted\ carbon\ (ppbC)\ =\ \Delta[VOC]\ (ppb)\ \times\ n_C$$

*where $\Delta[VOC]$ is the decrease in the VOC concentration during reaction with OH, and $n_C$ is the number of carbon atoms per molecule (8 for m-xylene, 10 for naphthalene).*

*Second, the carbon content of all observed products was calculated by summing the contribution of each compound:*

$$product\ carbon\ (ppbC)\ =\ \sum_{i}([product_i]\ (ppb)\ \times\ n_{C,i})$$

*where [product_i] is the measured mixing ratio of the product i and $n_{C,i}$ is the carbon number of each*
*product i.*

*The carbon balance was then computed as:*

$$carbon\ balance\ (\%)\ =\ \frac{product\ carbon\ (ppbC)}{reacted\ carbon\ (ppbC)}\ \times\ 100$$

*This calculation assumes that all carbon is either retained in the measured gas-phase or particle-*
*phase products or converted into undetected species. It does not account for carbon in*
*undetected products, such as CO, $CO_2$, and glyoxal that we could not measure because of*
*instrumental limitations."*

12. Line 267-269: Does increasing temperature promote the reactivity and ring cleavage of C8
compounds, thereby increasing the signal strength of C3 and C5 compounds?

The rate constant for m-xylene + OH increases by approximately 3.7 % between 280 K and 295 K,
based on Arrhenius parameters reported in the NIST Chemical Kinetics Database (activation
energy = -0.96 kJ mol$^{-1}$). This change falls within the typical uncertainty range of kinetic
measurements and is unlikely to meaningfully alter the oxidation dynamics. Therefore, while
increased OH concentrations at 295 K likely contribute to enhanced consumption of larger
products (like $C_6$ and $C_8$ species), the rate constant variation itself does not appear to be a
dominant factor in the observed shift toward smaller $C_3$ and $C_5$ products. Instead, it's more likely
that the increase in OH levels enhances fragmentation pathways, which explains the rise in $C_3$
and $C_5$ products. Finally, temperature is probably more affecting the partitioning more than
reactivity or thermal degradation.

13. Lines 461 – 465: How did the author come up with that there is no volatility trend as a function
of OSc for anthropogenic VOCs? Where is the plot shown in the manuscript?

The plot illustrating the relationship between OSc and volatility ($log_{10}C_i^*$) is provided in the SI as
Fig. S6, and also shown below. In this figure, OSc (y-axis) is plotted against volatility (x-axis),
where no clear correlation is observed between those two parameters.

[Figure]

**Figure S6. The OSc (y-axis) for the detected parent ions of (a) m-xylene and (b) naphthalene photooxidation, respectively, as a function of saturation concentration ($\log_{10}C_i^*$ in µg m$^{-3}$; x-axis) for high NO$_x$ experiments. The size of the circles denotes the mass of each species. Experiments carried out at 280 K are in light violet while the ones carried out at 295 K are in magenta.**

14. Figure 5: Are (c) and (d) only for the compound detected both in gas and particle phases? The same question applies to Figure 6.

Yes, Figures 5c, 5d and 6 include only compounds that are detected in both the gas and particle phases. This restriction is necessary because calculations of the partitioning coefficient ($K_p$) and the corresponding saturation concentration ($C_i^*$), as defined in Equations (2) and (3), require reliable measurements in both phases.

Technical Comment

1. Line 46: There is only one time when PAHs are used. You can use its full name.

The term PAH has been removed and we now directly refer to "naphthalene".

2. Line 47: It is unclear what is $C_1$

The "C1 position" refers to the α-carbon adjacent to the fusion of the two aromatic rings. This has been edited in the revised manuscript *(Introduction, Lines 48-49)*:

*"… at the α-carbon (68 %) adjacent to the fusion of its two aromatic rings (Wang et al., 2007)."*

3.   Line 61: "alkoxy radicals ($RO_2$, $HO_2$ and HO)". $RO_2$, $HO_2$ and HO were not alkoxy radicals.
Change to "peroxy radical ($RO_2$ and $HO_2$)".

We apologize for the wrong definition and have revised the text accordingly to use the correct
terminology: *"peroxy ($RO_2$ and $HO_2$) and OH radicals"*.

4.   Section 2.2 is very long. You can try to slice it into two or more paragraphs for readability.

Section 2.2 has been divided into shorter paragraphs.

5.   Lines 215: What is the meaning of the first steps of the experiment?

Here we meant the first phases of SOA formation corresponding to condensation or nucleation,
depending on experimental conditions. To avoid confusion, we have rephrased the sentence as
*(Section 3.1, Line 264-265)*: *"The absence of seed particles reduces the available surface area for*
*condensation of oxidized products."* This revised wording clarifies the intended meaning in the
manuscript.

6.   Figure 1: The authors try to summarise the literature data. It is a good attempt. However, due
to the small point sizes of this study and the use of multiple colours for different studies, I
found that it is hard to pinpoint the values from this study without careful reading. I would
suggest using one colour but different shapes for literature and keeping the two chosen
colours for this study and open and filled markers for high and low $NO_x$ The marker size should
be tuned to some extent for better readability. In addition, there is no need to scale the data
point by OH exposure because the authors did not discuss it in detail in the main text.

Figure 2 has been revised accordingly. We now use different marker shapes for literature data and
maintain filled versus open markers (with consistent colors) to distinguish between our study's
high- and low-NOx conditions. The scaling by OH exposure has been removed, and marker sizes
have been slightly increased to improve readability.

[Figure]

**Figure 2. SOA yields at 295 K and 280 K as function of organic aerosol mass formed for (a) m-xylene and (b)**
**naphthalene in comparison with previous studies. Filled markers correspond to high $NO_x$ conditions, open markers to**
**low $NO_x$.**

7.   Line 270: "Fig S3" should be "Fig S2".

The reference has been kept as Fig. S3 since a new figure has been added as Fig. S2.

8.   Figure S2: What is the left and right y-axis for?

The right axis in Fig. S3 (previously denoted as Fig. S2) corresponds to the concentration (ppb) of
$C_8H_{10}O_5$ (*m/z* 187.06) and is used for scaling to enable clearer visualization.

9.  Figure 5: Please label m-xylene and naphthalene in the plot and 280 K for the (c) and (d) plots.

Figure 5 has been updated and also presented below. We have now clearly labeled m-xylene and
naphthalene in panels (a) and (b), and added "at 280 K" labels to panels (c) and (d) for clarity.

[Figure]

**Figure 5. Panels (a) and (b) present the 2D-VBS framework: the O/C ratio (y-axis) for the detected parent ions of m-**
**xylene and naphthalene photooxidation, respectively, as a function of saturation concentration ($\log_{10}C_i^*$ in μg m⁻³; x-**
**axis) for high $NO_x$ experiments. The size of the circles denotes the mass of each species. Experiments carried out at 280**
**K are in light violet while the ones carried out at 295 K are in magenta. The light green and the light blue background**
**shadings correspond to the SVOC ($0 < \log_{10}C_i^* < 2.5$) and the IVOC ($2.5 < \log_{10}C_i^* < 5$) regimes, respectively. Panels (c)**
**and (d) present $\Delta\log_{10}C_i^*$ (the difference of $\log_{10}C_i^*$ between experiments at 280 and 295 K; y-axis) as function of carbon**
**number (x-axis), color scaled by oxygen number and sized by the mass (in μg m⁻³) of each compound at 280 K for m-**
**xylene and naphthalene oxidation products, respectively.**

10. Line 445: Is it supposed to be ug m⁻³?

This was a typo. It has been corrected to "μg m⁻³".

11. "Fig." and "Figure" should be used consistently.

According to ACP guidelines: "Figure" is used at the beginning of a sentence, and "Fig." is used
within the sentence. We have followed this convention throughout the manuscript.

https://www.atmospheric-chemistry-and-physics.net/submission.html